# Bernstein Polynomial-Based Method for Solving Optimal Trajectory Generation Problems

**DOI:** 10.3390/s22051869

**Published:** 2022-02-27

**Authors:** Calvin Kielas-Jensen, Venanzio Cichella, Thomas Berry, Isaac Kaminer, Claire Walton, Antonio Pascoal

**Affiliations:** 1Cooperative Autonomous Systems (CAS) Lab, Department of Mechanical Engineering, University of Iowa, Iowa City, IA 52242, USA; venanzio-cichella@uiowa.edu; 2Laboratory of Robotics and Engineering Systems (LARSyS), ISR/IST, University of Lisbon, 1049-001 Lisbon, Portugal; thomasdpberry@tecnico.ulisboa.pt (T.B.); antonio@isr.tecnico.ulisboa.pt (A.P.); 3Department of Mechanical and Aerospace Engineering, Naval Postgraduate School, Monterey, CA 93943, USA; kaminer@nps.edu; 4Department of Electrical Engineering, University of Texas at San Antonio, San Antonio, TX 78249, USA; claire.walton@utsa.edu

**Keywords:** optimal trajectory generation, Bernstein polynomials, Bézier curves, optimal control

## Abstract

This paper presents a method for the generation of trajectories for autonomous system operations. The proposed method is based on the use of Bernstein polynomial approximations to transcribe infinite dimensional optimization problems into nonlinear programming problems. These, in turn, can be solved using off-the-shelf optimization solvers. The main motivation for this approach is that Bernstein polynomials possess favorable geometric properties and yield computationally efficient algorithms that enable a trajectory planner to efficiently evaluate and enforce constraints along the vehicles’ trajectories, including maximum speed and angular rates as well as minimum distance between trajectories and between the vehicles and obstacles. By virtue of these properties and algorithms, feasibility and safety constraints typically imposed on autonomous vehicle operations can be enforced and guaranteed independently of the order of the polynomials. To support the use of the proposed method we introduce BeBOT (Bernstein/Bézier Optimal Trajectories), an open-source toolbox that implements the operations and algorithms for Bernstein polynomials. We show that BeBOT can be used to efficiently generate feasible and collision-free trajectories for single and multiple vehicles, and can be deployed for real-time safety critical applications in complex environments.

## 1. Introduction

The field of autonomous guidance has exploded in the past decade. Significant progress has been made in self driving vehicles, bringing them one step closer to reality [1]. Precision agriculture utilizes autonomous aerial vehicles to monitor crops and spray pesticides [2], and the development in autonomous weed pulling robots may reduce or eliminate the need for potentially harmful pesticides [3]. Underactuated marine surface vehicles can be controlled using a flatness-based approach [4]. Companies such as Amazon, Starship, and Zipline have already begun making autonomous deliveries [5,6,7]. In fact the first autonomous aerial vehicle has already been flown on a different planet [8]. This progress has led to high demand for computationally efficient algorithms that may yield safe and optimal trajectories to be planned for groups of autonomous vehicles. Our proposed method aims to accomplish these tasks by formulating the optimal trajectory generation problem as a nonlinear programming problem and exploiting the useful features of Bernstein polynomials.

Most techniques for planning and control of autonomous systems fall into one of two categories: closed-loop methods or open-loop methods. Closed-loop methods, sometimes referred to as feedback or reactive methods, use the current state knowledge to determine, in real time, what the next control input should be. On the other hand, open-loop methods determine control values or compute motion trajectories out to a specified time horizon with the use of the system’s model.

One common closed-loop technique that originally stemmed from maze solving algorithms is the bug algorithm. The bug algorithm, e.g., [9,10], uses local knowledge of the environment and a global goal to either follow a wall or move in a straight line towards the goal. This algorithm can be implemented on very simple devices due to typically requiring only two tactile sensors. However, it does not account for a vehicle’s dynamics and constraints. Moreover, bug algorithms are non-optimal methods and cannot be used for the execution of complex missions that require the optimization of some cost. For a review and comparison of bug algorithms, the reader is referred to [11].

Rather than working on an agent’s positions, the velocity obstacle (VO) algorithm uses relative velocities between the agent and obstacles to determine trajectories which will avoid collisions. The original term velocity obstacle was presented in [12]. Variations on the VO method include Common Velocity Obstacle [13], Nonlinear Velocity Obstacles [14], and Generalized Velocity Obstacles [15]. Other relevant closed-loop methods use artificial potential fields, which leverage a potential function providing attractive forces towards the goal and repulsive forces away from obstacles [16,17,18].

Among the advantages of closed-loop methods are fast computation and the ability to react to changing environments and unforeseen events. Furthermore, theoretical tools aimed at deriving safety guarantees of closed-loop methods are fairly well developed, and are mostly rooted in nonlinear systems analysis and robust and adaptive control. Despite these benefits, closed-loop methods are difficult to employ for multiple vehicle teams. They also generally lack the capability of presenting a human operator with a predicted trajectory and act rather like a black box, which can result in a lack of trust between the operator and the autonomous system.

In contrast to closed-loop methods, open-loop methods can generate solutions in one-shot for the whole mission time, and are therefore able to present an operator with an intuitive representation of the future trajectory. This representation is typically shown as a 2D or 3D path and may also include speed, acceleration, and higher derivatives of the vehicle’s motion. Randomized algorithms such as probabilistic roadmaps (PRMs) [19] and rapidly exploring random trees (RRT, RRT*) [20,21] randomly sample the work space to reduce computational complexity. PRMs randomly sample feasible regions within the work space to construct a dense graph. A graph-based solver can then be used to determine the optimal route. RRTs compute trajectories by using directed sampling to build trees. This approach can find feasible solutions in situations involving both a high number of constraints and high dimensional search spaces. Unfortunately, random sampling algorithms may be difficult to use in real-time applications due to computational complexity and may end up exploring regions that will not lead to a solution.

Similar to PRMs, other graph-based approaches aim to efficiently build and then search a graph. Cell decomposition methods, e.g., [22,23], build a graph of their environment by recursively increasing the resolution of areas of interest resulting in a few large nodes of open space and many small nodes near obstacles. Once a graph has been built, a graph solver can be used. A popular graph solver is the A* algorithm [24], which is an extension of Dijkstra’s algorithm that uses a heuristic function to improve the search speed. Many modifications to the A* algorithm also exist, such as Lifelong Planning A* [25], which replans a path anytime an obstacle appears on the existing path and utilizes Dijkstra’s algorithm to transition the robot from its current pose to the new path. Similarly, Anytime Dynamic A* (ADA*) [26] iteratively decreases a suboptimality bound to improve the plan’s optimality within a specified maximum computation time limit. Iterative approaches such as ADA*, sometimes called “anytime” methods, compute a coarse solution and then refine it until a computation timeout is reached. For example, Ref. [27] investigates the addition of committed trajectories and branch-and-bound tree adaptation to the RRT* algorithm to produce an online anytime method.

In addition to graph-based representations of trajectories, polynomial approximation methods can be used as well. In [28], trajectories are represented as piecewise polynomial functions and are generated in a manner that minimizes their snap. In TrajOpt [29], a sequential quadratic program is solved to generate optimal polynomial trajectories while performing continuous time collision checking.

Other open-loop methods include CHOMP [30,31], STOMP [32], and HOOP [33]. In CHOMP, infeasible trajectories are pulled out of collisions while simultaneously smoothing the trajectories using covariant gradient descent. STOMP adopts a similar cost function to that found in CHOMP, but generalizes to cost functions whose gradients are not available. This is done by stochastically sampling noisy trajectories. HOOP utilizes a problem formulation which computes vehicle trajectories in two steps. In the first step, a path is planned quickly by considering only the vehicle’s kinematics. The second step then refines this trajectory into a higher order piecewise polynomial using a quadratic program. Other methods, such as WASPAS-mGqNS [34], balance the optimality of motion plans with respect to the mission objectives against exploring unknown environments.

Open-loop methods provide useful tools for dealing with high dimensional problems such as multiple vehicles and several constraints. They are also capable of producing trajectories that accomplish multiple goals. However, due to the curse of dimensionality, the computational complexity of open-loop methods grows significantly with the number of vehicles, constraints, and goals. For the most part, motion planning methods trade optimality and/or safety for computational speed. Our goal is to introduce a method that mitigates this trade-off, and that provides provably safe solutions for high dimensional problems while retaining the computational efficiency of low-order trajectory planning algorithms. This is achieved by exploiting the useful features of Bernstein polynomials.

The Bernstein basis was originally introduced by Sergei Natanovich Bernstein (1880–1968) in order to provide a constructive proof of Weierstrass’s theorem. Bernstein polynomials were not widely used until the advent of digital computers due to their slow convergence as function approximants. Widespread adoption eventually occurred when it was realized that the coefficients of Bernstein polynomials could be intuitively manipulated to change the shape of curves described by these polynomials. In the 1960s, two French automotive engineers became interested in using this idea: Paul de Faget de Casteljau and Pierre Étienne Bézier.

Designing complex shapes for automobile bodies by sculpting clay models proved to be a time consuming and expensive process. To combat this, de Casteljau and Bézier sought to develop mathematical tools that would allow designers to intuitively construct and manipulate complex shapes. Due to de Casteljau publishing most of his research internally at his place of employment, Bézier’s name became more widely associated with Bernstein polynomials, frequently referred to as Bézier curves. Building on existing research and modern technology, Bernstein polynomials provide several useful properties for many fields.

The Bernstein basis provides numerical stability [35], as well as useful geometric properties and computationally efficient algorithms that can be used to derive and implement efficient algorithms for the computation of trajectory bounds, trajectory extrema, minimum temporal and spatial separation between two trajectories and between trajectories and obstacles, and collision detection. Bernstein polynomials also allow for the representation of continuous time trajectories using low-order approximations.

Our method for trajectory generation builds upon [36,37,38], where Bernstein polynomials were introduced as a tool to approximate the solutions of nonlinear optimal control problems with provable convergence guarantees. While the results concerning the convergence of the Bernstein approximation method are out of the scope of the present paper, here we focus on the design of algorithms and functions for Bernstein polynomials. These include evaluating bounds, minimum spatial distance, collision detection, and penetration distance. Additionally, we show how these properties can be used for trajectory generation in realistic mission scenarios such as trajectory generation for swarms, navigating cluttered environments, and motion planning for vehicles operating in a Traveling Salesman mission. For the interested reader, an open-source implementation is provided. This paper extends the results initially presented in [39]. In particular, in [39] we focused on autonomous vehicle trajectories representation by Bernstein polynomials, and proposed a preliminary implementation of BeBOT for minimum distance computation and collision detection for safe autonomous operation. In the present paper, we extend previous work by exploiting properties and proposing algorithms for both Bernstein polynomials and rational Bernstein polynomials. BeBOT includes an open-source Python implementation of these algorithms, which enables the user to exploit the properties of (rational) Bernstein polynomials for trajectory generation. Furthermore, we address several applications for multiple autonomous systems and show the efficacy of BeBOT in enabling safe autonomous operations. We added a new algorithm, the penetration algorithm, and several additional examples including air traffic control, navigation of a cluttered environment, vehicle overtaking, 1000 vehicle swarming, a marine vehicle example, and two examples of a vehicle routing problem. An implementation of the examples presented in this paper is available at our GitHub website [40] and can be customized to facilitate the toolbox’s usability.

The goal of this manuscript is to provide a general framework which can be applied to a plethora of different systems ranging from mobile robots to manipulators. However, we do provide several numerical examples for mobile robots and include the governing motion equations for ease of implementation, e.g., see examples in Section 5.1 and Section 5.6.

In brief, the main contributions of this article are:Novel algorithms which exploit the useful properties of (rational) Bernstein polynomials for use in trajectory generation.Several examples implementing the aforementioned algorithms in realistic mission scenarios.

The paper is structured as follows. In Section 2 we introduce Bernstein polynomials and their properties. Section 3 introduces the use of Bernstein polynomials to parameterize 2D and 3D trajectories. In Section 4 we present computationally efficient algorithms for the computation of state and input constraints typical of trajectory generation applications. In Section 5 we demonstrate the efficacy of these algorithms through several numerical examples. The paper ends with Section 6, which draws some conclusions. A Python implementation of the properties and algorithms presented, as well as the scripts used to generate the plots and examples found throughout this paper, can be found on our GitHub webpage [40].

In what follows, vectors are denoted by bold letters, e.g., $p={\left[p_{x}\;,\;p_{y}\right]}^{⊤}$ and ||·|| denotes the Euclidean norm (or magnitude), e.g., $\left||p|\right|=\sqrt{p_{x}^{2}+p_{y}^{2}}$.

## 2. Mathematical Preliminaries

The motion planning problems addressed in this work can be in general formulated as optimal control problems. Letting the states and control inputs of the vehicles be denoted by x(t) and u(t), respectively, the optimal motion planning problem can formally be stated as follows:(1)$$\begin{array}{cc} \; & \min_{x(t),u(t)}I\left(x\left(t\right),u\left(t\right)\right)=E\left(x\left(0\right),x\left(t_{f}\right)\right)+\int_{0}^{t_{f}}F\left(x\left(t\right),u\left(t\right)\right)dt\; \end{array}$$
subject to
(2)$$\begin{array}{cc} \; & \dot{x}\left(t\right)=f\left(x\left(t\right),u\left(t\right)\right)\;,\;\forall t\in \left[0,t_{f}\right], \end{array}$$
(3)$$\begin{array}{cc} \; & e(x\left(0\right),x\left(t_{f}\right))=0\;, \end{array}$$
(4)$$\begin{array}{cc} \; & h\left(x\left(t\right),u\left(t\right)\right)\leq 0\;,\;\forall t\in \left[0,t_{f}\right]\;, \end{array}$$
where $I:R^{n_{x}}\times R^{n_{u}}\to R$, $E:R^{n_{x}}\times R^{n_{x}}\to R$, $F:R^{n_{x}}\times R^{n_{u}}\to R$, $f:R^{n_{x}}\times R^{n_{u}}\to R^{n_{x}}$, $e:R^{n_{x}}\times R^{n_{x}}\to R^{n_{e}}$, and $h:R^{n_{x}}\times R^{n_{u}}\to R^{n_{h}}$.

Here, *I* defined in Equation (Equation 1) is a Bolza-type cost functional, with end point cost *E* and running cost *F*. The constraint in Equation (Equation 2) enforces the dynamics of the vehicles considered, Equation (Equation 3) enforces the boundary conditions, e.g., initial and final position, speed, heading angles of the vehicles, and Equation (Equation 4) describes feasibility and mission specific constraints, e.g., minimum and maximum speed, acceleration, collision avoidance constraints, etc.

In previous work [36,37,38] we presented a discretization method to approximate state and input by *n*th order Bernstein polynomials. This approximation allows us to transcribe the optimal control problem into a non-linear programming problem, which can then be solved by off-the-shelf optimization solvers. In particular, we show that the solution to the non-linear programming problem converges to the solution of the original optimal control problem as *n* increases. The present paper focuses on the geometric properties of Bernstein polynomials and their implementation for computationally efficient and safe trajectory generation. In the following, we report the properties of Bernstein polynomials and rational Bernstein polynomials which are relevant to this paper.

An *n*th order Bernstein polynomial defined over an arbitrary interval $\left[t_{0},t_{f}\right]$ is given by
(5)$$C_{n}\left(t\right)=\sum_{i=0}^{n}P_{i,n}B_{i,n}\left(t\right),\;t\in \left[t_{0},t_{f}\right]\;,$$
where $P_{i,n}\in R^{D}$ is the *i*th Bernstein coefficient, *D* is the number of dimensions, and $B_{i,n}\left(t\right)$ is the Bernstein polynomial basis defined as
Bi,n(t)=ni(t−t0)i(tf−t)n−i(tf−t0)n,ni=n!i!(n−i)!,
for all i=0,…,n. Typically the dimensionality of a Bernstein polynomial, *D*, is either 2 or 3 for 2D or 3D spatial curves, respectively. In this case, Bernstein polynomials are often referred to as Bézier curves and their Bernstein coefficients are known as *control points*. While Bézier’s original work did not explicitly use the Bernstein basis [41,42], it was later shown that the original formulation is equivalent to the Bernstein form polynomial [43].

An *n*th order *rational* Bernstein polynomial, $R_{n}\left(t\right)$, is defined as
(6)$$R_{n}\left(t\right)=\frac{\sum_{i=0}^{n}P_{i,n}w_{i,n}B_{i,n}\left(t\right)}{\sum_{i=0}^{n}w_{i,n}B_{i,n}\left(t\right)},\;t\in \left[t_{0},t_{f}\right],$$
where $w_{i,n}\in R$, i=0,…,n, are referred to as weights. A list of relevant properties of Bernstein polynomials used throughout this article can be found in Appendix A.

## 3. Generation of 2D and 3D Trajectories Using (Rational) Bernstein Polynomials

### 3.1. 2D Trajectories

Here we will examine several illustrative examples of the properties of Bernstein polynomials and rational Bernstein polynomials in 2D. All the plots presented can be generated using the example code available at [40].

Figure 1, Figure 2, Figure 3, Figure 4, Figure 5, Figure 6, Figure 7, Figure 8 and Figure 9 contain several examples of 2D trajectories in the spatial domain. Two trajectories are plotted in Figure 1 along with an obstacle. The trajectories $C^{\left[1\right]}\left(t\right)$ and $C^{\left[2\right]}\left(t\right)$ are defined as in Equation (Equation 5) with $t_{0}=10$ s and $t_{f}=20$ s where the Bernstein coefficients are temporally equidistant. The vector of Bernstein coefficients of trajectory $C^{\left[1\right]}\left(t\right)$ is
$$P_{5}^{\left[1\right]}=\left[\begin{array}{cccccc} 0 & 2 & 4 & 6 & 8 & 10\\ 5 & 0 & 2 & 3 & 10 & 3 \end{array}\right].$$

The vector of Bernstein coefficients of trajectory $C^{\left[2\right]}\left(t\right)$ is
$$P_{5}^{\left[2\right]}=\left[\begin{array}{cccccc} 1 & 3 & 6 & 8 & 10 & 12\\ 6 & 9 & 10 & 11 & 8 & 8 \end{array}\right].$$

The circular obstacle has a radius of 1 and is centered at point ${\left[3,4\right]}^{⊤}$. Figure 2 highlights the endpoints property (Property A2). Note that the trajectory $C^{\left[1\right]}\left(t\right)$ passes through its first and last Bernstein coefficients $P_{0,5}^{\left[1\right]}={\left[0,5\right]}^{⊤}$ and ${\left[10,3\right]}^{⊤}$, respectively. Likewise, the trajectory $C^{\left[2\right]}\left(t\right)$ passes through its first and last Bernstein coefficients ${\left[1,6\right]}^{⊤}$ and ${\left[12,8\right]}^{⊤}$, respectively. The convex hull property (Property A1) is illustrated in Figure 3.

Useful operations can be efficiently performed on Bernstein polynomials by manipulating only their coefficients. The de Casteljau algorithm (Property A5) allows one to split a Bernstein polynomial into two separate polynomials. This is shown in Figure 4 where trajectories $C^{\left[1\right]}\left(t\right)$ and $C^{\left[2\right]}\left(t\right)$ are split at $t_{\mathrm{div}}=15$ s. Degree elevation (Property A6) is performed on both trajectories in Figure 5. Note that in both Figure 4 and Figure 5, the convex hulls are more accurate than the conservative convex hulls in Figure 3. This idea will be expanded upon in the next section.

Bernstein polynomials can also be used to extract useful information about the dynamics of the trajectories. In Figure 6, Bernstein polynomials representing the squared speed of trajectories $C^{\left[1\right]}\left(t\right)={\left[x^{\left[1\right]}\left(t\right),y^{\left[1\right]}\left(t\right)\right]}^{⊤}$ and $C^{\left[2\right]}\left(t\right)={\left[x^{\left[2\right]}\left(t\right),y^{\left[2\right]}\left(t\right)\right]}^{⊤}$ are shown along with their corresponding coefficients and convex hulls. The squared speed is computed using the derivative and arithmetic operation properties (Properties A3 and A7) as follows
$${\left(v^{\left[1\right]}\left(t\right)\right)}^{2}={\left({\dot{x}}^{\left[1\right]}\left(t\right)\right)}^{2}+{\left({\dot{y}}^{\left[1\right]}\left(t\right)\right)}^{2}$$

Note that the squared speed of a trajectory described by a Bernstein polynomial is also a Bernstein polynomial.

Letting $C^{\left[1\right]}\left(t\right)={\left[x^{\left[1\right]}\left(t\right)\;,\;y^{\left[1\right]}\left(t\right)\right]}^{⊤}$, the heading angle ψ(t) of a trajectory can be computed as
(7)$$\psi^{\left[1\right]}\left(t\right)=\tan^{-1}\left(\frac{{\dot{y}}^{\left[1\right]}\left(t\right)}{{\dot{x}}^{\left[1\right]}\left(t\right)}\right).$$

Since the inverse tangent of a Bernstein polynomial is not a Bernstein polynomial, we take the tangent of both sides of the equation, yielding
(8)$$\tan\left(\psi^{\left[1\right]}\left(t\right)\right)=\left(\frac{{\dot{y}}^{\left[1\right]}\left(t\right)}{{\dot{x}}^{\left[1\right]}\left(t\right)}\right),$$
which is a rational Bernstein polynomial. Figure 7 illustrates the quantity expressed in Equation (Equation 8) computed for trajectories $C^{\left[1\right]}\left(t\right)$ and $C^{\left[2\right]}\left(t\right)$.

To determine the angular rate along the trajectory at any point in $t\in [t_{0},t_{f}]$, we can take the derivative of the heading angle, yielding
(9)$$\omega^{\left[1\right]}\left(t\right)={\dot{\psi }}^{\left[1\right]}\left(t\right)=\frac{{\dot{x}}^{\left[1\right]}\left(t\right){\ddot{y}}^{\left[1\right]}\left(t\right)-{\ddot{x}}^{\left[1\right]}\left(t\right){\dot{y}}^{\left[1\right]}\left(t\right)}{{\left({\dot{x}}^{\left[1\right]}\left(t\right)\right)}^{2}+{\left({\dot{y}}^{\left[1\right]}\left(t\right)\right)}^{2}}.$$

Since the angular rate can be determined using Properties A3 and A7, it can be represented as a rational Bernstein polynomial. The angular rates of trajectories $C^{\left[1\right]}\left(t\right)$ and $C^{\left[2\right]}\left(t\right)$ are shown in Figure 8.

Finally, the Bernstein polynomial representing the squared distance between two trajectories at every point in time can be computed from
(10)$$\begin{array}{cc} d^{2}\left(t\right) & ={\left(x^{\left[2\right]}\left(t\right)-x^{\left[1\right]}\left(t\right)\right)}^{2}+{\left(y^{\left[2\right]}\left(t\right)-y^{\left[1\right]}\left(t\right)\right)}^{2},\\ \; & \forall t\in [t_{0},t_{f}] \end{array}$$

The center point of a circular, static obstacle Obs(t), can be represented as a Bernstein polynomial whose coefficients are all identical and set to the position of the obstacle, i.e.,
$$P^{\left[Obs\right]}=\left[\begin{array}{ccc} x^{\left[Obs\right]} & \cdots & x^{\left[Obs\right]}\\ y^{\left[Obs\right]} & \cdots & y^{\left[Obs\right]} \end{array}\right].$$

The degree of the Bernstein polynomial representing the center point of the circular obstacle is equal to that of the order of the Bernstein polynomials representing the trajectories.

### 3.2. 3D Trajectories

We now introduce two 3D Bernstein polynomials with $t_{0}=10$ s and $t_{f}=20$ s, where the coefficients are equidistant in time, and illustrate their properties in Figure 10, Figure 11, Figure 12, Figure 13, Figure 14, Figure 15, Figure 16 and Figure 17. The Bernstein coefficients of trajectory $C^{\left[3\right]}\left(t\right)$ are
$$P_{5}^{\left[3\right]}=\left[\begin{array}{cccccc} 7 & 3 & 1 & 1 & 3 & 7\\ 1 & 2 & 3 & 8 & 3 & 5\\ 0 & 2 & 1 & 9 & 8 & 10 \end{array}\right],$$
and the Bernstein coefficients of trajectory $C^{\left[4\right]}\left(t\right)$ are
$$P_{5}^{\left[4\right]}=\left[\begin{array}{cccccc} 1 & 1 & 4 & 4 & 8 & 8\\ 5 & 6 & 9 & 10 & 8 & 6\\ 1 & 1 & 3 & 5 & 11 & 6 \end{array}\right].$$

These polynomials are drawn in Figure 10.

Similar to the 2D examples, Figure 11, Figure 12, Figure 13 and Figure 14 illustrate the end points, convex hull, de Casteljau, and elevation properties, respectively. Figure 15 and Figure 16 show the squared speed and squared acceleration of trajectories $C^{\left[3\right]}\left(t\right)$ and $C^{\left[4\right]}\left(t\right)$, respectively. These values were computed using the derivative and arithmetic properties. Finally, Figure 17 shows the squared Euclidean distance between the trajectories and the center of the spherical obstacle at every point in time. The distance was found using Property A7.

## 4. Algorithms for (Rational) Bernstein Polynomials

This section contains algorithms and procedures for Bernstein polynomials that make use of the properties presented in Section 2. These functions include: evaluating bounds, using the convex hull property to quickly find conservative bounds; evaluating extrema, through an iterative procedure that computes a solution within a desired tolerance; minimum spatial distance, applying a similar iterative procedure to find the minimum spatial distance between two Bernstein polynomials; and collision detection, which quickly determines whether a collision may exist or does not exist.

### 4.1. Evaluating Bounds

Property A1 allows one to quickly establish conservative bounds on the Bernstein polynomial. For example, given the 2D Bernstein polynomial, see Equation (Equation 5), with coefficients given by
$$P_{5}=\left[\begin{array}{cccccc} 0 & 1 & 2 & 3 & 4 & 5\\ 5 & 0 & 2 & 5 & 7 & 5 \end{array}\right]\;,$$
lower and upper bounds $C_{\min}$ and $C_{\max}$ satisfying $C_{\min}\leq C\left(t\right)\leq C_{\max},\;\forall t\in \left[t_{0},t_{f}\right]$ can be derived using Equation (Equation 25). Figure 18 exhibits the Bernstein polynomial (solid blue line) given the above coefficients (orange dots connected with dashes). The most conservative estimate of the minimum and maximum *Y* values of the Bernstein polynomial is given by the coefficients with the lowest and highest *Y* values, respectively. The lower bound is 0 and the upper bound is 7. As mentioned in Property A6 and Equation (Equation 32), the Bernstein coefficients converge to the curve as the polynomial is degree elevated. This fact can be used to derive tighter bounds. A degree elevation of 20 results in a lower bound of 1.93 and an upper bound of 5.89. This is a closer estimate of the actual minimum and maximum, 2.26 and 5.70, respectively (see red dotted lines and Section 4.2). Figure 18 also illustrates degree elevations of 5, 10, and 15. Since the degree elevation matrix, see Equation (Equation 31), is independent of the Bernstein coefficients, a database of elevation matrices can be computed ahead of time to produce tight estimates of the bounds at a low computational cost.

### 4.2. Evaluating Extrema

The extrema of a Bernstein polynomial are calculated using an iterative procedure similar to the one proposed in [44]. This is done by recursively splitting the curve and using the Convex Hull (Property A1) to obtain an estimate within some desired tolerance. Algorithm 1 outlines the process for determining the maximum of a Bernstein polynomial and can easily be modified to determine the minimum of a Bernstein polynomial.

The inputs to Algorithm 1 are the Bernstein polynomial’s coefficients, $P=\{P_{n}\}$, $P_{n}=\left[P_{0,n},\ldots ,P_{n,n}\right]$, an arbitrarily large *negative* global lower bound, α, and a desired tolerance, ϵ. Note that in order to compute a reliable maximum, α≤min(P). In practice, α is set to the lowest possible value that can be reliably represented in the computer.

Line 1 finds the maximum of the two endpoints of the Bernstein polynomial, where *n* is the degree of the polynomial. This makes use of the End Points (Property A2). Next, we determine the upper bound by simply finding the maximum of P. The *if* statement on line 3 determines whether the global lower bound should be replaced with the current lower bound. The next *if* statement, line 6, will prune the current set of Bernstein coefficients. This is valid because α always provides a lower bound on the global maximum. If the upper bound of any subset is below α, then we know that it is impossible for any point on that subset to be the global maximum. The final *if* statement, line 9, determines whether the difference between the upper and lower bounds is within the desired tolerance and returns the global minimum bound α if the tolerance is met.

The *else* statement, starting on line 11, splits the curve and then recursively calls Algorithm 1 again. The function split() utilizes the de Casteljau algorithm (Property A5). One of two different splitting points, $t_{\mathrm{div}}$, can be employed. The first option simply splits the curve in half, i.e., $t_{\mathrm{div}}=t_{0}+\frac{t_{f}-t_{0}}{2}$. The second option uses the index of the largest valued coefficient, $i_{ub}=\mathrm{argmax}\left(P\right)$, to determine the splitting point, i.e., $t_{\mathrm{div}}=t_{0}+\left(t_{f}-t_{0}\right)\frac{i_{ub}}{n}$.

Algorithm 1 (and its converse) is employed to find the minimum and maximum of the 5th degree Bernstein polynomial depicted in Figure 18 (red lines). The execution time to compute the minimum is 320 μs on a Lenovo ThinkPad laptop using an Intel Core i7-8550U, 1.80 GHz CPU. The implementation can be found in [40].    

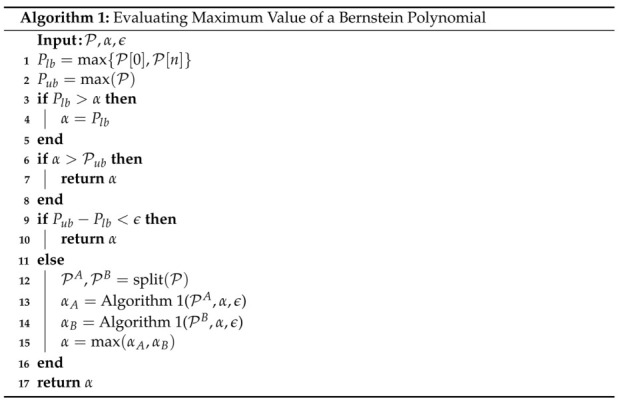


### 4.3. Minimum Spatial Distance

The minimum spatial distance between two Bernstein polynomials can be computed using the method outlined in [44]. This is done by exploiting the Convex Hull property (Property A1), the End Point Values property (Property A2), the de Casteljau Algorithm (Property A5), and the Gilbert-Johnson-Keerthi (GJK) algorithm [45]. The latter is widely used in computer graphics to compute the minimum distance between convex shapes.

The algorithm for the minimum spatial distance between two Bernstein polynomials is shown in Algorithm 2. The first two inputs to the function are the sets of Bernstein coefficients, $P=\{P_{m}\}$ and $Q=\{Q_{n}\}$, which define the two Bernstein polynomials in question. The last two inputs are the global upper bound on the minimum distance, α, and a desired tolerance ϵ.

The upper_bound() function on line 1 finds the furthest distance between the end points of the two Bernstein polynomials, i.e., lower_bound(P,Q)=max{P[0]−Q[0],P[0]−Q[n],P[m]−Q[0],P[m]−Q[n]} where *m* and *n* are the degrees of the polynomials represented by P and Q, respectively. This is a valid upper bound on the minimum distance between the two polynomials due to End Point Values (Property A2).

The lower_bound() function on line 2 finds the lower bound on the distance between the two polynomials by using the GJK algorithm. This is a valid lower bound because of Property A1, Convex Hull. The next three *if* statements on lines 3, 6, and 9 are very similar to those seen in Algorithm 1. Line 3 updates the global upper bound α if the current upper bound is smaller. Line 6 prunes the current iteration since it is impossible the current lower bound, lower, to be the minimum distance if it is larger than the global upper bound. Line 9 returns α if the desired tolerance is met.

The lines within the *else* statement split the Bernstein polynomials defined by P and Q and recursively call Algorithm 2. Like in Algorithm 1, the first option for splitting would be to simply split at the halfway point. The second option for splitting the curves is outlined in [44] and uses the location at which the minimum distance occurs to choose the splitting point. Figure 19a illustrates the minimum distance between several different Bernstein polynomials. The code to generate this plot can be found in [40]. The execution time to compute the minimum spatial distance is 3.29 ms on a Lenovo ThinkPad laptop using an Intel Core i7-8550U, 1.80 GHz CPU.

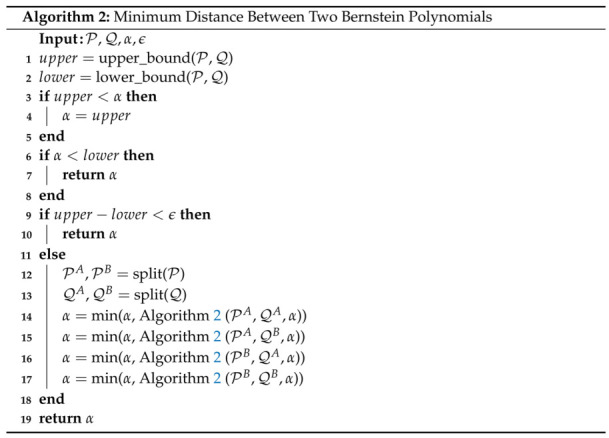


**Remark** **1.**
*Note that Algorithm 2 can also be employed to compute the minimum distance between a Bernstein polynomial and a point or a convex shape. This is shown in Figure 19b.*


### 4.4. Collision Detection

In some cases it may be desirable to quickly check the feasibility of a trajectory rather than finding a minimum distance. The collision detection algorithm can be used in these cases. The two major differences between the Collision Detection Algorithm and the Minimum Distance Algorithm described previously are a modification of the GJK algorithm and a change in the stopping criteria. Rather than having the GJK algorithm return a minimum distance, it simply returns whether a collision has been detected (i.e., convex hulls intersecting). The stopping criteria is set to return the moment no collisions are found rather than continuing iterations to meet a desired tolerance. For example, if the original convex hulls of two Bernstein polynomials do not intersect, the collision detection algorithm will return *no collision* after the first iteration while the minimum distance algorithm will continue to iterate until the desired tolerance is met. Therefore, this algorithm is computationally inexpensive compared to the minimum distance algorithm, with the drawback that it only returns a binary value (no collision or collision possible) rather than a minimum distance.

The collision detection algorithm is shown in Algorithm 3. The inputs are the coefficients of the Bernstein polynomials being compared, P and Q, and the maximum number of iterations max_iter. The *while* loop beginning on line 2 runs until it is determined that a collision does not exist or until the maximum number of iterations is met. The find_collisions() function on line 3 uses the modified GJK algorithm to determine which convex hulls from the set P collide with those from the set Q. The *if* statement on line 4 checks to see whether collisions were found. If both $P_{col}$ and $Q_{col}$ are empty, then no collisions exist. If collisions do exist then the *for* loops starting on lines 7 and 11 split all the convex hulls that were found to collide and add them to the set to be checked. Note that the parent set that is split is removed from the set of convex hulls to check. If the maximum number of iterations is met, then the algorithm returns that a collision is possible. The execution time when a collision is possible is 1.10 ms on a Lenovo ThinkPad laptop using an Intel Core i7-8550U, 1.80 GHz CPU. However, when a collision does not exist, the execution time is only 7.25 μs.

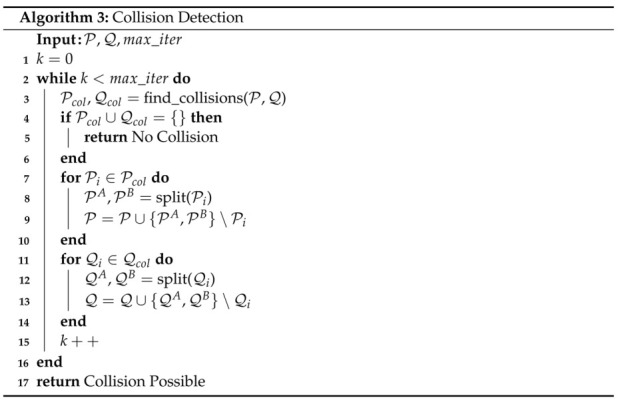



### 4.5. Penetration Algorithm

If two convex shapes intersect, in order to derive information such as the penetration depth and vector, the EPA [46] can be used. A slight modification of the EPA algorithm is proposed here, which is referred to as the DEPA, whose objective is to find the penetration of one convex shape relative to another along a specific direction $\vec{d}$. The top left plot of Figure 20 shows two shapes intersecting each other, and the remaining plots show examples of penetration vectors, i.e., the vector $\vec{d}$ needed to move the second shape so that it no longer intersects the first shape. The DEPA algorithm finds the shortest possible penetration vector. The pseudocode is reported below (see Algorithm 4).

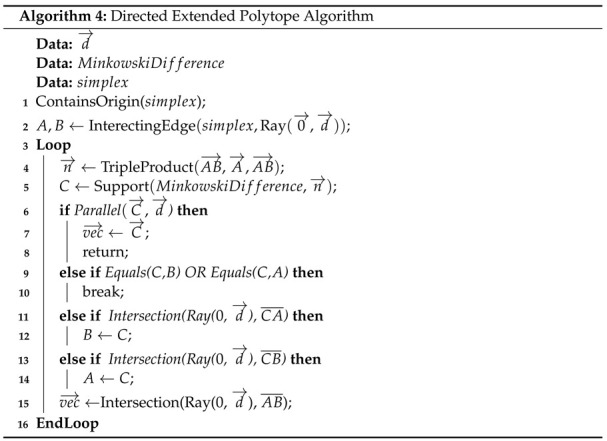


Figure 21 demonstrates the algorithm in 4 steps. The first plot shows the Minkowski Difference of the shapes depicted in Figure 22, which contains the origin and with a triangle simplex that contains the origin. This is the desired direction to move a polytope A (blue polytope) of Figure 22 such that it no longer contains polytope B (beige polytope). Once the norm of the point along the edge of the Minkowski Difference parallel to $\vec{d}$ is found, A can then move in the same direction with the same length to no longer intersect B.

## 5. Numerical Examples

In this section, numerical examples using the BeBOT toolkit and Python’s Scipy Optimization package are examined (flight tests are available at [47]). The implementation of the following examples can be found in [40].

### 5.1. Dubins Car—Time Optimal

In this simple example, several trajectories for a vehicle with Dubins car dynamics are generated to illustrate the properties of Bernstein polynomials. We let the desired trajectory assigned to the vehicle be given by the Bernstein polynomial
(11)$$\left[\begin{array}{c} C_{n}^{\left[x\right]}\left(t\right)\\ C_{n}^{\left[y\right]}\left(t\right) \end{array}\right]=C_{n}\left(t\right)=\sum_{i=0}^{n}P_{i,n}B_{i,n}\left(t\right),\;t\in \left[t_{0},t_{f}\right].$$

The square of the speed of the vehicle is a 1D Bernstein polynomial given by
$$v^{2}\left(t\right)=\left|\right|{\dot{C}}_{n}\left(t\right){\left|\right|}^{2}.$$

The heading angle is
(12)$$\psi \left(t\right)=\tan^{-1}\frac{{\dot{C}}_{n}^{\left[y\right]}\left(t\right)}{{\dot{C}}_{n}^{\left[x\right]}\left(t\right)},$$
and the angular rate is a 1D rational Bernstein polynomial given by
(13)$$\omega \left(t\right)=\frac{{\ddot{C}}_{n}^{\left[y\right]}\left(t\right){\dot{C}}_{n}^{\left[x\right]}\left(t\right)-{\dot{C}}_{n}^{\left[y\right]}\left(t\right){\ddot{C}}_{n}^{\left[x\right]}\left(t\right)}{\left|\right|{\dot{C}}_{n}\left(t\right){\left|\right|}^{2}}.$$

The objective at hand is to find a trajectory that arrives at a desired destination in the minimum possible time while adhering to feasibility and safety constraints. In particular, the trajectory generation problem is as follows:$$\begin{array}{cc} \; & \min_{P_{n},t_{f}}t_{f} \end{array}$$
subject to
$$\begin{array}{cc} \; & C_{n}\left(t_{0}\right)=C_{0},\;C_{n}\left(t_{f}\right)=C_{f},\\ \; & \psi \left(t_{0}\right)=\psi_{0},\;\psi \left(t_{f}\right)=\psi_{f}\\ \; & \left|\right|{\dot{C}}_{n}\left(t_{0}\right)\left|\right|=v_{0},\;||{\dot{C}}_{n}\left(t_{f}\right)\left|\right|=v_{f},\\ \; & \left|\right|{\dot{C}}_{n}\left(t\right){\left|\right|}^{2}\leq v_{\max}^{2},\;\forall t\in \left[t_{0},t_{f}\right]\\ \; & \left|\right|\dot{\psi }\left(t\right)\left|\right|\leq \omega_{\max},\;\forall t\in \left[t_{0},t_{f}\right]\\ \; & \left|\right|C_{n}\left(t\right)-O_{i}{\left|\right|}^{2}\geq d_{s}^{2},\;\forall t\in \left[t_{0},t_{f}\right],\;i=1,2. \end{array}$$

We set the initial and final position, heading, and speed to $C_{0}={\left[3,0\right]}^{⊤}$ m, $C_{f}={\left[7,10\right]}^{⊤}$ m, $\psi_{0}=\psi_{f}=\frac{\pi }{2}$ rad, and $v_{0}=v_{f}=1\;\frac{m}{s}$. The maximum speed, maximum angular rate, and minimum safe distance constraints are $v_{\max}=5\;\frac{m}{s}$, $\omega_{\max}=1\frac{\mathrm{rad}}{s}$, and $d_{s}^{2}=1$ m, respectively. The positions of the obstacles are $O_{1}={\left[3,2\right]}^{⊤}$ m and $O_{2}={\left[6,7\right]}^{⊤}$ m.

In the problem above, the initial and final constraints for position, heading, and speed are enforced using the End Point Values property (Property A2) together with Equations (Equation 26), (Equation 12) and (Equation 13). Similarly, the same property is used to enforce the initial and final speeds and headings (see (Equation 26)). Note that the norm squared of the speed and of the distance between the trajectory and the obstacles can be expressed as Bernstein polynomials (the sum, the difference, and the product between Bernstein polynomials are also Bernstein polynomials). A similar argument can be made for the norm square of the angular rate, which can be expressed as a rational Bernstein polynomial (see Property A7). Thus, the maximum speed and angular rate, and collision avoidance constraints can be enforced using the Evaluating Bounds or Evaluating Extrema procedures described in Section 4.1 and Section 4.2.

Figure 23 shows the results with n=10. The blue curve is obtained by enforcing the constraints using the Evaluating Bounds procedure. The optimal time of arrival at the final destination is $t_{f}=9.14$ s. Next, we solve the same problem by enforcing the constraints using the Evaluating Bound procedure together with Degree Elevation. Recall that by degree elevating a Bernstein polynomial, the Bernstein coefficients converge towards the curve. Thus, degree elevation can be used to enforce constraints with tighter bounds. The orange and green lines show the trajectories obtained using degree elevations of 30 and 100, respectively. Degree elevation to degree 30 results in an optimal final time $t_{f}=7.64$ s. The elevation to degree 100 provides an optimal value $t_{f}=7.12$ s. Finally, the trajectory with smallest optimal final time, $t_{f}=6.45$ s, depicted as the red curve in Figure 23, is obtained by enforcing the constraints using the Evaluating Extrema algorithm (Section 4.2). While higher degree elevations or evaluating the exact extrema can produce more optimal trajectories, that optimality comes at the cost of additional computation time. Using a Lenovo Thinkpad P52s with an Intel Core i7-8550U CPU with a 1.8 GHz clock and 8 GB of memory, the computation time required for no degree elevation, a degree elevation of 30, a degree elevation of 100, and the exact extrema algorithm was 0.105 s, 0.146 s, 0.201 s, and 0.573 s, respectively.

Figure 24 illustrates the squared speed of each example. Figure 25 shows the angular rate of each trial. It can be seen that the vehicle correctly adheres to the speed and angular rate constraints for each trial with the only differences being the final time and proximity to the obstacles.

**Remark** **2.**
*The Exact Extrema function is a complex non-linear and non-smooth function. When it is used to enforce constraints, gradient-based optimization solvers such as the one used in this work can fail to converge to a feasible solution, especially if the initial guess is not feasible. One option is to use an iterative procedure where (i) a feasible sub-optimal solution is obtained by enforcing the collision avoidance constraint using the Evaluating Bounds function, and (ii) this solution is then used as an initial guess to solve the (more accurate) problem with the Exact Extrema constraint.*


### 5.2. Air Traffic Control—Time Optimal

In this example, we consider the problem of routing several commercial flights between major US cities in two dimensions (i.e., constant altitude). Assuming that each flight departs at the same time, the goal is to minimize the combined flight time of all the vehicles. Let the position, speed, heading, and angular rate of each vehicle under consideration be parameterized as in Section 5.1. We shall also make the assumption that the trajectories are on a 2D plane rather than on the surface of the Earth.

The goal is to compute cumulatively time optimal trajectories subject to maximum speed and angular velocity bounds, initial and final position, angle, and speeds. The vehicles must also maintain a minimum safe distance between each other. This problem can be formulated as follows:$$\min_{P_{n},t_{f}}\sum_{k=1}^{m}t_{f}^{\left[k\right]}$$
subject to
$$\begin{array}{cc} \; & C_{n}^{\left[k\right]}\left(0\right)=C_{0}^{\left[k\right]},\;C_{n}^{\left[k\right]}\left(t_{f}^{\left[k\right]}\right)=C_{f}^{\left[k\right]},\\ \; & \psi^{\left[k\right]}\left(0\right)=\psi_{0}^{\left[k\right]},\;\psi^{\left[k\right]}\left(t_{f}^{\left[k\right]}\right)=\psi_{f}^{\left[k\right]},\\ \; & \left|\right|{\dot{C}}_{n}^{\left[k\right]}\left(0\right)||=v_{0}^{\left[k\right]},\;||{\dot{C}}_{n}^{\left[k\right]}\left(t_{f}^{\left[k\right]}\right)\left|\right|=v_{f}^{\left[k\right]},\\ \; & v_{\min}^{2}\leq \left|\right|{\dot{C}}_{n}^{\left[k\right]}\left(t\right){\left|\right|}^{2}\leq v_{\max}^{2},\;\forall t\in \left[0,t_{f}^{\left[k\right]}\right],\\ \; & |{\dot{\psi }}^{\left[k\right]}\left(t\right)|\leq \omega_{\max},\;\forall t\in \left[0,t_{f}^{\left[k\right]}\right],\\ \; & \left|\right|C_{n}^{i}\left(t\right)-C_{n}^{j}\left(t\right){\left|\right|}^{2}\geq d_{s}^{2},\;\forall i,j\in \left\{1,\ldots ,m\right\},i\neq j. \end{array}$$
where the superscript [k] corresponds to the *k*th vehicle out of *m* vehicles, $C_{0}^{\left[k\right]}$ and $C_{f}^{\left[k\right]}$ are the initial and final positions, $\psi_{0}^{\left[k\right]}$ and $\psi_{f}^{\left[k\right]}$ are the initial and final headings, $v_{0}^{\left[k\right]}$ and $v_{f}^{\left[k\right]}$ are the initial and final speeds, $v_{\min}$ and $v_{\max}$ are the minimum and maximum speeds, $\omega_{\max}$ is the maximum angular velocity, $d_{s}$ is the minimum safe distance, and $t_{f}^{\left[k\right]}$ is the final time of the *k*th vehicle.

The departure cities, in vehicle order, are: San Diego, New York, Minneapolis, and Seattle. The arrival cities, in vehicle order, are: Minneapolis, Seattle, Miami, and Denver. The initial and final speeds are all $v_{0}^{\left[k\right]}=v_{f}^{\left[k\right]}=205\frac{m}{s}\;\forall k\in \left\{1,\ldots ,m\right\}$, the initial headings are $\psi_{0}={\left[0,\pi ,0,0\right]}^{⊤}$ rad, the final headings are $\psi_{f}=\left[0,\pi ,-\frac{\pi }{2},0\right]$ rad, the minimum speed is $v_{\min}=200\;\frac{m}{s}$, the maximum speed is $v_{\max}=260\;\frac{m}{s}$, the maximum angular velocity is $\omega_{\max}=3\frac{\deg}{s}=0.0524\frac{\mathrm{rad}}{s}$, the minimum safe distance is $d_{s}=5$ km, and the degree of the Bernstein polynomials being used is 5.

The initial and final position constraints are enforced using the End Point Values property (Property A2). Similarly, the same property is used to enforce the initial and final speeds and headings (see (Equation 26)). Note that the norm square of the speed and the norm square of the distance between vehicles can be expressed as 1D Bernstein polynomials (the sum, difference, and product between Bernstein polynomials are also Bernstein polynomials). A similar argument can be made for the norm square of the angular rate, which can be expressed as a rational Bernstein polynomial (see Property A7). Thus, the maximum speed and angular rate, and collision avoidance constraints can be enforced using the Evaluating Bounds or Evaluating Extrema procedures described in Section 4.1 and Section 4.2.

The optimized flight plans can be seen in Figure 26. The squared speed of each vehicle is shown in Figure 27. Note that each vehicle begins and ends with the same speed. The vehicles never slow down less than their initial speeds which means they never reach the minimum speed constrain, nor do the vehicles go faster than the maximum speed. In Figure 28, the angular velocity of each vehicle is shown. The minimum and maximum angular rate constraints are shown by the dotted lines. The vehicles’ angular rates never approach the minimum or maximum angular rate constraints due to the large area being covered. Finally, the squared euclidean distance between vehicles is shown in Figure 29. As expected, the squared Euclidean distance between two vehicles never falls below the minimum safe distance. Note that curves within the constraint plots end at different times. This is expected since each vehicle has a different final time. The furthest time reached in Figure 29 is less than that of the other plots because the other vehicles have already reached their final time before the longest flight reaches its final time.

### 5.3. Cluttered Environment

In many real world scenarios, robots must safely traverse cluttered environments. In this example, three aerial vehicles traveling at a constant altitude must navigate around several obstacles while also adhering to dynamic and minimum safe distance constraints. Let the position, speed, heading angle, and angular rate of each vehicle be defined as in Section 5.1. The goal of this example is to compute trajectories whose arc length is minimized subject to maximum speed constraints along with initial and final positions, heading angles, and speeds. The vehicles should also adhere to a minimum safe distance between each other and between obstacles. We formulate the problem as follows:(14)$$\min_{P_{n}}\sum_{i=1}^{m}\sum_{k=0}^{n-1}\left|\right|P_{k+1,n}^{\left[i\right]}-P_{k,n}^{\left[i\right]}\left|\right|$$
subject to
$$\begin{array}{cc} \; & C_{n}^{\left[k\right]}\left(0\right)=C_{0}^{\left[k\right]},\;C_{n}^{\left[k\right]}\left(t_{f}\right)=C_{f}^{\left[k\right]},\\ \; & \psi^{\left[k\right]}\left(0\right)=\psi_{0}^{\left[k\right]},\;\psi^{\left[k\right]}\left(t_{f}\right)=\psi_{f}^{\left[k\right]},\\ \; & \left|\right|{\dot{C}}_{n}^{\left[k\right]}\left(0\right)||=v_{0}^{\left[k\right]},\;||{\dot{C}}_{n}^{\left[k\right]}\left(t_{f}\right)\left|\right|=v_{f}^{\left[k\right]},\\ \; & \left|\right|{\dot{C}}_{n}^{\left[k\right]}\left(t\right){\left|\right|}^{2}\leq v_{\max}^{2},\;\forall t\in \left[0,t_{f}\right],\\ \; & \left|\right|C_{n}^{\left[i\right]}\left(t\right)-C_{n}^{\left[j\right]}\left(t\right){\left|\right|}^{2}\geq d_{s}^{2},\;\forall i,j\in \left\{1,\ldots ,m\right\},i\neq j,\\ \; & \left|\right|C_{n}^{\left[i\right]}\left(t\right)-O_{j}{\left|\right|}^{2}\geq d_{obs}^{2},\;\forall t\in \left[0,t_{f}\right],\;i\in \left\{1,\ldots ,m\right\},\\ \; & \;\;j\in \{1,\ldots ,b\}, \end{array}$$
where $O_{j}$ is the position of the *j*th obstacle out of *b* obstacles.

The initial positions for each vehicle, in order, are ${\left[0,0\right]}^{⊤}$ m, ${\left[10,0\right]}^{⊤}$ m, and ${\left[20,0\right]}^{⊤}$ m. The initial speeds are all $1\;\frac{m}{s}$ and the initial heading angles are all $\frac{\pi }{2}$ rad. The final positions for each vehicle are, in order, ${\left[20,30\right]}^{⊤}$ m, ${\left[0,30\right]}^{⊤}$ m, and ${\left[10,30\right]}^{⊤}$ m. The final speeds and final heading angles are the same as the initial speeds and heading angles. The order of the Bernstein polynomials being used is 7, the final time is $t_{f}=30$ s, the minimum safe distance between vehicles is $d_{s}=1$ m, the minimum safe distance between vehicles and obstacles is $d_{obs}=2$ m, and the maximum speed is $v_{\max}=10\;\frac{m}{s}$. The vehicles traversing the cluttered environment can be seen in Figure 30. This experiment has been repeated in the Cooperative Autonomous Systems (CAS) lab using three AR Drones 2.0. The flight tests can be viewed at [47].

### 5.4. Vehicle Overtake

Here we consider an autonomous driving example in which one vehicle attempts to overtake another vehicle while driving around a $90^{\circ }$ corner. The corner is defined by two arcs with a center point located at ${\left[140,0\right]}^{⊤}$ m. The inner track has a radius of $r_{inner}=125$ m and the outer track has a radius of $r_{outer}=140$ m. To clearly distinguish the vehicle being overtaken, it will be referred to as the opponent.

For simplicity, we consider the objective of minimizing the arc-length of the trajectory, which can be done by minimizing the sum of the squared Euclidean norm of consecutive control points, i.e.,
(15)$$E\left(P_{n}\right)=\sum_{i=1}^{n}\left|\right|P_{i,n}-P_{i-1,n}{\left|\right|}^{2}.$$

The desired endpoint of the vehicle is at the end of the corner. This is computed by measuring the angle between the vehicle’s position and the end of the curve,
(16)$$A\left(P_{n}\right)={\left(\mathrm{argtan}2\left(P_{n,n}^{\left[y\right]}-q^{\left[y\right]},P_{n,n}^{\left[x\right]}-q^{\left[x\right]}\right)-\frac{\pi }{2}\right)}^{2},$$
where the function argtan2 returns an angle in the correct quadrant [48]. Given Equations (Equation 15) and (Equation 16), we formulate the problem as
(17)$$\min_{P_{n}}\left(\left(1-\alpha \right)E\left(P_{n}\right)+\alpha A\left(P_{n}\right)\right)\beta$$
subject to
$$\begin{array}{cc} \; & C_{n}\left(0\right)=C_{0},\;\psi \left(0\right)=\psi_{0},\;||{\dot{C}}_{n}\left(0\right)\left|\right|=v_{0},\\ \; & \left|\right|{\dot{C}}_{n}\left(t\right){\left|\right|}^{2}\leq v_{\max}^{2},\;\forall t\in \left[0,t_{f}\right],\\ \; & |\dot{\psi }\left(t\right)|\leq \omega_{\max},\;\forall t\in \left[0,t_{f}\right],\\ \; & r_{inner}^{2}\leq \left|\right|C_{n}\left(t\right)-q{\left|\right|}^{2}\leq r_{outer}^{2},\\ \; & \left|\right|C_{n}\left(t\right)-O_{n}\left(t\right){\left|\right|}^{2}\geq d_{s}^{2}, \end{array}$$
where α and β are tuning parameters. $C_{0}$, $\psi_{0}$, and $v_{0}$ are the initial position, heading, and speed of the vehicle, respectively. $v_{\max}$ and $\omega_{\max}$ are the maximum speed and angular rate, respectively. The predicted trajectory of the opponent is represented as the Bernstein polynomial $O_{n}\left(t\right)$ and the minimum safe distance to the opponent is $d_{s}$. Using a sensor such as a camera or LiDAR, one could measure the state of the opponent and then predict its future position using a method such as the one presented in [49].

At time $t=t_{0}$, when planning occurs, the position of the vehicle is ${\left[5,0\right]}^{⊤}$ m, its speed is $50\;\;\frac{m}{s}$, and its initial heading angle is $\frac{\pi }{2}\;\mathrm{rad}$. The control points of the Bernstein polynomial representing the opponent’s trajectory are
$$\left[\begin{array}{cccc} 10 & 25 & 50 & 70\\ 10 & 75 & 90 & 105 \end{array}\right]m.$$

The maximum speed is $65\;\frac{m}{s}$, the maximum angular rate is $\frac{\pi }{5}\;\frac{\mathrm{rad}}{s}$, and the minimum safe distance is 3 m. The tuning parameters are $\alpha =1-10^{-6}$ and β=100.

Figure 31 illustrates the optimized vehicle trajectory overtaking the opponent’s trajectory. Figure 32 shows the squared speed of the vehicle along with the maximum speed constraint. As expected, the vehicle’s speed approaches the maximum speed in order to successfully overtake the opponent. Figure 33 shows the vehicle’s angular rate and its upper and lower constraints. It is clear that the vehicle remains within the desired bounds. Figure 34 shows the squared distance between the vehicle and the opponent. While the vehicle does come close to the opponent, it is never closer than the minimum safe distance.

### 5.5. Swarming

This section examines two methods for generating trajectories for large groups of autonomous aerial vehicles. The centralized method optimizes every trajectory at once. On the other hand, the decentralized method generates trajectories one at a time and compares them to previously generated trajectories.

The position of each vehicle in a swarm of *m* vehicles for the following examples is parameterized as a 3D Bernstein polynomial, i.e.,
$$\sum_{i=0}^{n}P_{i,n}^{\left[j\right]}B_{i,n}\left(t\right)=C_{n}^{\left[j\right]}\left(t\right),\;\forall j\in \left\{1,\ldots ,m\right\},\;P_{n}^{\left[j\right]}\in R^{3\times n}.$$

#### 5.5.1. 101 Vehicle—Centralized

The centralized method optimizes the trajectories for each vehicle simultaneously. The goal is to minimize the arc length of each trajectory. There are *m* vehicles with 3rd order Bernstein polynomials representing their trajectories which are constrained to a minimum safe distance between each other and initial and final positions. This is formulated as follows: $$\min_{P_{n}}\sum_{i=1}^{m}\sum_{k=0}^{n-1}\left|\right|P_{k+1}^{\left[i\right]}-P_{k}^{\left[i\right]}\left|\right|,$$
subject to
$$\begin{array}{cc} \; & C_{n}^{\left[i\right]}\left(0\right)=C_{0}^{i},\;C_{n}^{\left[i\right]}\left(t_{f}\right)=C_{f}^{\left[i\right]},\;\forall i\in \left\{1,\ldots ,m\right\},\\ \; & \left|\right|C_{n}^{\left[i\right]}\left(t\right)-C_{n}^{\left[j\right]}\left(t\right){\left|\right|}^{2}\geq d_{s}^{2},\;\forall i,j\in \left\{1,\ldots ,m\right\},i\neq j. \end{array}$$

The initial positions for each vehicle were chosen randomly from a 25m×25 m grid at an altitude of z=0 m. The final positions were chosen to spell out “CAS”, as seen in Figure 35, at an altitude of z=100 m. In the next section we significantly reduce the number of dimensions in the optimization vector by using the decentralized approach.

#### 5.5.2. 101 Vehicle—Decentralized

The decentralized method iteratively computes trajectories for the *i*th vehicle. Each new iteration is compared to the previously computed trajectories so that the minimum safety distance constraint is met. The problem that is solved at each iteration is written as
$$\min_{P_{n}^{\left[i\right]}}\sum_{i=1}^{m}\sum_{k=0}^{n-1}\left|\right|P_{k+1}^{\left[i\right]}-P_{k}^{\left[i\right]}\left|\right|$$
subject to
$$\begin{array}{cc} \; & C_{n}^{\left[i\right]}\left(0\right)=C_{0}^{\left[i\right]},\;C_{n}^{\left[i\right]}\left(t_{f}\right)=C_{f}^{\left[i\right]},\\ \; & \left|\right|C_{n}^{\left[i\right]}\left(t\right)-C_{n}^{\left[j\right]}\left(t\right){\left|\right|}^{2}\geq D_{s}^{2},\;\forall j\in \left\{1,\ldots ,i-1\right\},\;i>1. \end{array}$$

Note that the first vehicle does not need to satisfy the minimum safe distance constraint since no trajectories have been computed before it.

The parameters used in this example were identical to that of the previous subsection. The resulting figure has been omitted due to its similarity to Figure 35.

#### 5.5.3. 1000 Vehicle—Decentralized

The decentralized method can be used to compute 1000 trajectories. In this example, it is employed to generate the paths seen in Figure 36 to display the University of Iowa Hawkeye logo. The initial points are equally dispersed at an altitude of z=0m on a 100m×100m grid. The final points are the pattern shown at an altitude of z=100. The cost function aims to maximize the temporal distance between the current *i*th trajectory and the previously generated *j*th trajectories by taking the reciprocal of the sum of the Bernstein coefficients of the norm squared difference, i.e.
$$\min_{P_{n}^{\left[i\right]}}\frac{1}{\sum_{j=1}^{i-1}P^{\left[norm,j\right]}},\;i>1,$$
subject to
$$\begin{array}{cc} \; & C_{n}^{\left[i\right]}\left(0\right)=C_{0}^{\left[i\right]},\;C_{n}^{\left[i\right]}\left(t_{f}\right)=C_{f}^{\left[i\right]}, \end{array}$$
where $P^{\left[norm,j\right]}$ are the Bernstein coefficients of the Bernstein polynomial representing the squared temporal distance between the *i*th and *j*th trajectories, i.e.,
$$\left|\right|C^{\left[i\right]}\left(t\right)-C^{\left[j\right]}\left(t\right){\left|\right|}^{2}=\sum_{i=0}^{n}P^{\left[norm,j\right]}B_{i,n}\left(t\right).$$

It should be noted that this formulation of cost function and constraints is used as a proof of concept. For other possible cost function and constraint formulations, the reader is referred to [50,51].

### 5.6. Marine Vehicle Model

In this example, we consider a marine vehicle model known as the medusa. The equations of motion of the medusa are as follows
(18)$$\begin{array}{c} \dot{x}=u\cos\psi -v\sin\psi ,\\ \dot{y}=u\sin\psi +v\cos\psi ,\\ \dot{\psi }=r. \end{array}$$
(19)$$\begin{array}{c} m_{u}\dot{u}-m_{v}vr+d_{u}u=\tau_{u},\\ m_{v}\dot{v}+m_{u}ur+d_{v}v=0,\\ m_{r}\dot{r}-m_{uv}uv+d_{r}r=\tau_{r}, \end{array}$$
where *x* and *y* represent the vehicle’s position, ψ is the orientation, *u* (surge) and *v* (sway) are the linear velocities, *r* is the turning rate, and $\tau ={\left[\tau_{u},\tau_{r}\right]}^{T}$ is the vector of forces and torques due to thrusters/surfaces (control input).

In this example, we let the state, ${\left[x,y,\psi ,u,v,r\right]}^{⊤}$, and input, ${\left[\tau_{u},\tau_{r}\right]}^{⊤}$, be approximated by Bernstein polynomials, and impose the vehicle’s dynamics directly through Bernstein polynomial differentiation. Using Property A3, the dynamics constraints given by Equations (Equation 18) and (Equation 19) reduce to a set of algebraic constraints. Additional constraints imposed on this problem include collision avoidance and input saturation constraints. Figure 37 shows an example of motion planning for a medusa vehicle, which is required to reach a final destination in the minimum time. Ten markers are plotted along the trajectory (shown in blue) to represent the heading of the vehicle at that point in time. It can easily be seen that the vehicle’s trajectory avoids the (inflated) unsafe region illustrated by the orange circle.

### 5.7. Dynamic Routing Problem

#### 5.7.1. Single Vehicle Case

We consider a problem where a single vehicle is supposed to visit *M* neighborhoods $B_{i}=\{x\in R^{2}:∥x-b_{i}∥\leq r$ } in minimum time $t_{f}$. Here r>0 and the vectors $b_{i}\in R^{2}$, i∈[1,M] represent a sequence of points of interest that has been generated by a Traveling Salesman Problem (TSP) algorithm. Let $t_{i}$ denote a time instance when the vehicle’s position satisfies $C_{n}\left(t_{i}\right)\in B_{i}$. Then, the dynamic routing problem for a single vehicle can be formulated as follows,${\mathrm{DR}}_{1}$
(20)$$\min_{t_{i},i\in \left[1,M\right],\;\;P_{n}}t_{f}$$
subject to
$$\begin{array}{c} C_{n}\left(t_{i}\right)\in B_{i}\\ t_{i-1}<t_{i}<t_{i+1},\;\forall i=2,M-1\\ t_{1}>0,\;t_{f}>t_{M}\\ ∥{\dot{C}}_{n}{∥}_{\infty }\leq 1,\\ ∥{\ddot{C}}_{n}{∥}_{\infty }\leq 1,\\ C_{n}\left(0\right)=C_{n}\left(t_{f}\right)=C_{0} \end{array}$$

#### 5.7.2. Numerical Solution: Single Vehicle Case

Let
(21)$$C_{n}^{i}\left(t\right)=\left\{\begin{array}{c} \sum_{k=0}^{n}P_{i,k}B_{k,n}\left(t\right),t\in \left[t_{i-1},t_{i}\right],t_{0}=0,t_{M+1}=t_{f}\\ 0,\;o.w. \end{array}\right.,i\in \left[1,M+1\right]$$

Define
(22)$$C_{n}\left(t\right)=\sum_{i=1}^{M+1}C_{n}^{i}\left(t\right)$$

Then the numerical solution of the dynamic routing problem ${\mathrm{DR}}_{1}$ was obtained by solving the optimization problem
(23)$$\min_{t_{i},i\in \left[1,M\right],\;\;P_{n}^{i}}t_{f}$$
subject to
$$\begin{array}{c} ∥C_{n}^{i}\left(t_{i}\right)-b_{i}∥\leq r\\ t_{i-1}<t_{i}<t_{i+1},\;\forall i=2,M-1\\ t_{1}>0,\;t_{f}>t_{M}\\ ∥{\dot{C}}_{n}{∥}_{\infty }\leq 1,\\ ∥{\ddot{C}}_{n}{∥}_{\infty }\leq 1,\\ C_{n}\left(0\right)=C_{n}\left(t_{f}\right)=C_{0}\\ C_{n}^{i}\left(t_{i}\right)=C_{n}^{i+1}\left(t_{i}\right),\;i\in \left[1,M-1\right]\\ {\dot{C}}_{n}^{i}\left(t_{i}\right)={\dot{C}}_{n}^{i+1}\left(t_{i}\right)\;i\in \left[1,M-1\right] \end{array}$$

A simulation was performed illustrating a single agent visiting 30 neighborhoods. The resulting trajectory is shown in Figure 38. The agent is limited to velocities of arbitrary units ranging from −1 to 1 and is similarly limited to accelerations of arbitrary units also from −1 to 1. The velocities and accelerations of the vehicle can be seen in Figure 39 and Figure 40, respectively.

#### 5.7.3. Multiple Vehicle Case

In this case, *K* drones are assigned a total of *K* neighborhood sets to visit. Each neighborhood set, $P_{k}$, consists of an equal number of neighborhoods $B_{ij}$ which are defined by a set of points of interest $b_{ik}\in R^{2}$, i.e.,
$$B_{ik}=\{x\in R^{2}:∥x-b_{ik}∥\leq r,\;i\in \left[1,M\right],\;k\in \left[1,K\right]\},$$
and
$$P_{k}=\left\{B_{1k},\ldots \ldots ,B_{Mk},\;k\in \left[1,K\right]\right\}.$$

Let $t_{fk},k\in \left[1,K\right]$ denote the total time it takes for the *k*th vehicle to visit every neighborhood in the set $P_{k}$ once and let $t_{ik}$ denote a time instance when the *k*th vehicle’s position satisfies $C_{n}^{k}\left(t_{ik}\right)\in B_{ik}$. Using this notation, we propose the following definition of the multi-vehicle dynamic routing problem for given positive number $w_{k}$ and *d*${\mathrm{DR}}_{2}$(24)$$\min_{t_{ik},i\in \left[1,M\right],k\in \left[1,K\right],\;\;P_{n}^{k}}\;\;\sum_{k=1}^{K}w_{k}t_{fk}$$
subject to
$$\begin{array}{c} C_{n}^{k}\left(t_{ik}\right)\in B_{ik}\in P_{k},\forall k\in \left[1,K\right]\\ t_{i-1,k}<t_{ik}<t_{i+1,k},\;\forall i\in \left[2,M-1\right],k\in \left[1,K\right]\\ t_{1k}>0,\;t_{fk}>t_{Mk}\\ |{\ddot{C}}_{n}^{k}\left(t\right)|\leq u_{max},\;\forall t\in \left[0,t_{fk}\right]\\ C_{n}^{k}\left(0\right)=C_{n}^{k}\left(t_{f}\right)=C_{k0},\;\forall k\in \left[1,K\right]\\ |C_{n}^{k}\left(t\right)-C_{n}^{l}\left(t\right)|\geq d,\;k\neq l,\forall t\in \left[0,\max_{j\in [1,K]}\;t_{fj}\right],\;k\in \left[1,K\right],\;l\in \left[1,K\right]. \end{array}$$

Simulations were performed for a two agent and ten agent case each visiting 10 neighborhoods per agent. The resulting trajectories for the two agent case are shown in Figure 41 and for the ten agent case in Figure 42.

## 6. Conclusions

We presented a method to generate optimal trajectories by using Bernstein polynomials to transcribe the problem into a nonlinear programming problem. By exploiting the useful properties of Bernstein polynomials, our method provides computationally efficient algorithms that can also guarantee safety in continuous time which are useful in optimization routines. These algorithms include evaluating bounds, evaluating extrema, minimum spatial distance between two Bernstein polynomials, minimum spatial distance between a Bernstein polynomial and a convex polygon, collision detection, and the penetration algorithm. We also developed an open source toolbox which makes these transcription methods readily available in the Python programming language.

Numerical examples were provided to demonstrate the efficacy of the method. Simple cost functions and constraints were implemented to generate trajectories for several realistic mission scenarios including air traffic control, navigating a cluttered environment, overtaking a vehicle, trajectory generation for a large swarm of vehicles, trajectory generation for a marine vehicle, and navigation for vehicles operating in a Traveling Salesman mission. Our formulation offers a powerful tool for users to generate optimal trajectories in real time scenarios for single or multiple robot teams. Future work includes developing new cost functions, exploring different optimization frameworks, and replanning trajectories to react to a changing environment.

## Figures and Tables

**Figure 1 sensors-22-01869-f001:**
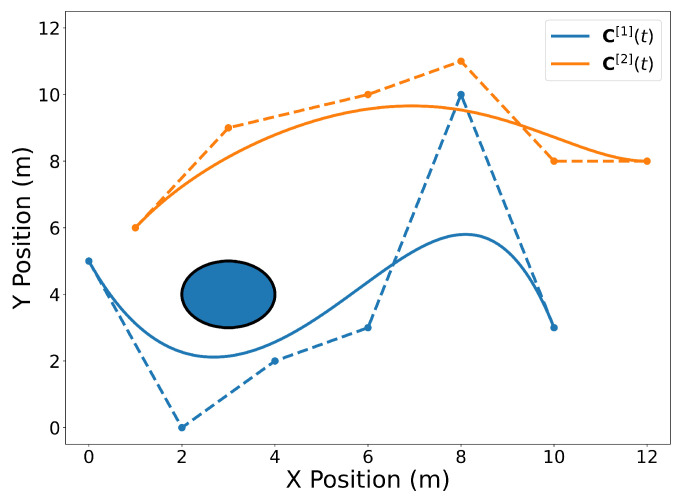
Spatial representation of two Bernstein polynomial trajectories in 2D near a circular obstacle. Trajectory $C^{\left[1\right]}\left(t\right)$ is drawn in blue and trajectory $C^{\left[2\right]}\left(t\right)$ is drawn in orange. The solid lines are the polynomials and the dashed lines connect the Bernstein coefficients for convenience. Note that the temporal aspect of the trajectories above has been omitted for clarity of presenting the geometric properties of Bernstein polynomials.

**Figure 2 sensors-22-01869-f002:**
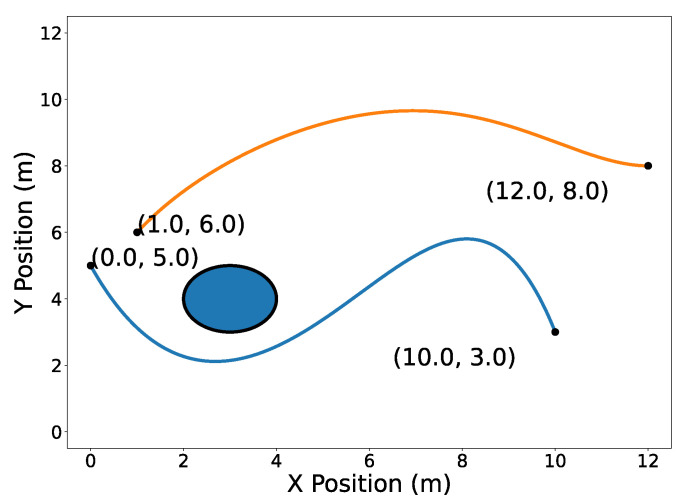
Trajectories $C^{\left[1\right]}\left(t\right)$ (blue) and $C^{\left[2\right]}\left(t\right)$ (orange) with their endpoints highlighted in 2D.

**Figure 3 sensors-22-01869-f003:**
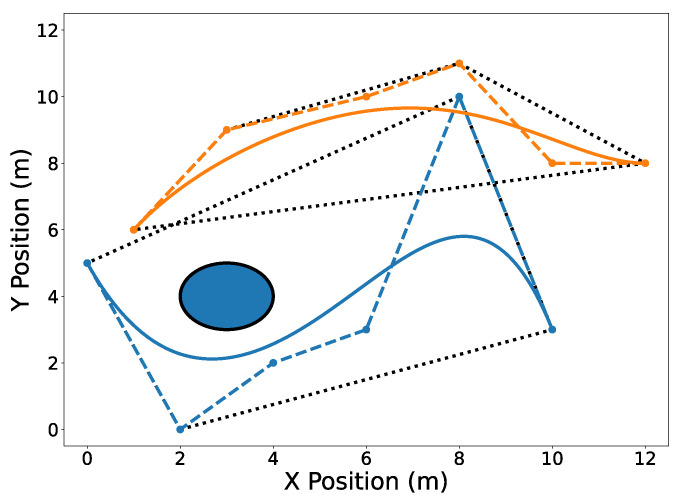
Convex hulls drawn as black dotted lines around the Bernstein coefficients of trajectories $C^{\left[1\right]}\left(t\right)$ (blue) and $C^{\left[2\right]}\left(t\right)$ (orange). The dashed lines connect the control points of their corresponding trajectories.

**Figure 4 sensors-22-01869-f004:**
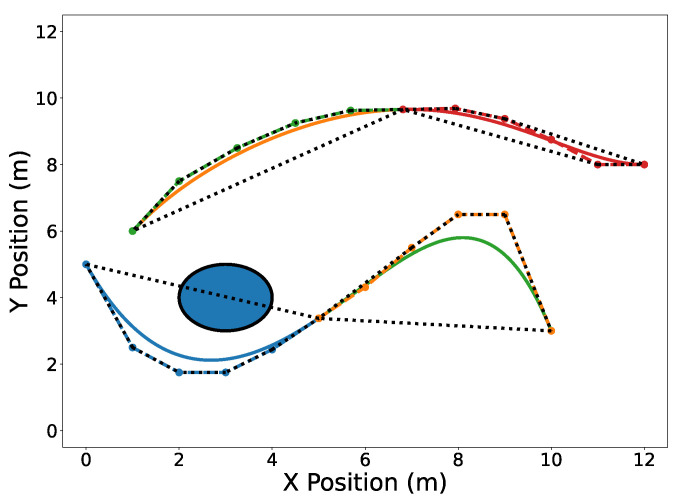
Trajectories $C^{\left[1\right]}\left(t\right)$ (split into blue and green) and $C^{\left[2\right]}\left(t\right)$ (split into orange and red) split at $t_{\mathrm{div}}=15$ s. Convex hulls are drawn around the Bernstein coefficients of the new split trajectories.

**Figure 5 sensors-22-01869-f005:**
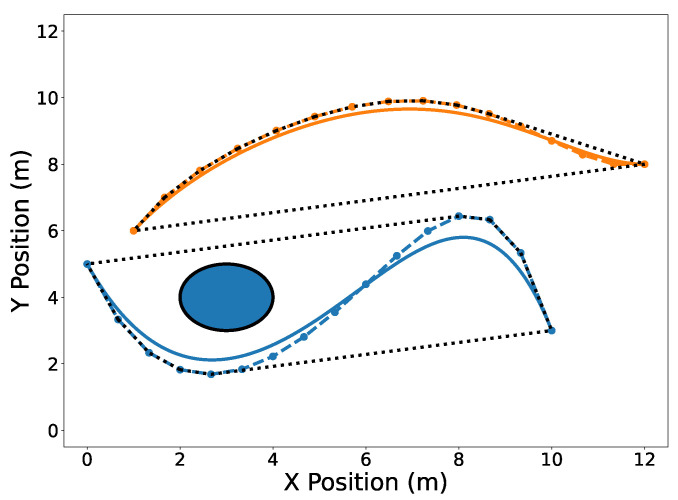
Convex hull drawn around the elevated Bernstein coefficients of trajectories $C^{\left[1\right]}\left(t\right)$ (blue) and $C^{\left[2\right]}\left(t\right)$ (orange).

**Figure 6 sensors-22-01869-f006:**
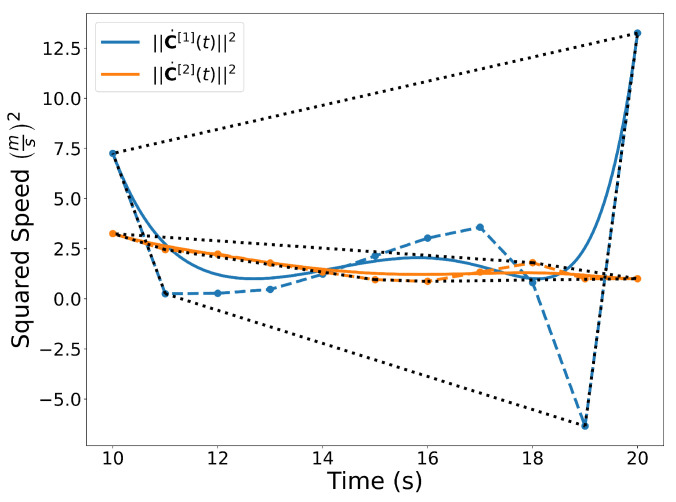
Squared speed of the trajectories $C^{\left[1\right]}\left(t\right)$ and $C^{\left[2\right]}\left(t\right)$. A convex hull is drawn around the Bernstein coefficients. Note that even though the coefficients may be negative, the actual curve is not.

**Figure 7 sensors-22-01869-f007:**
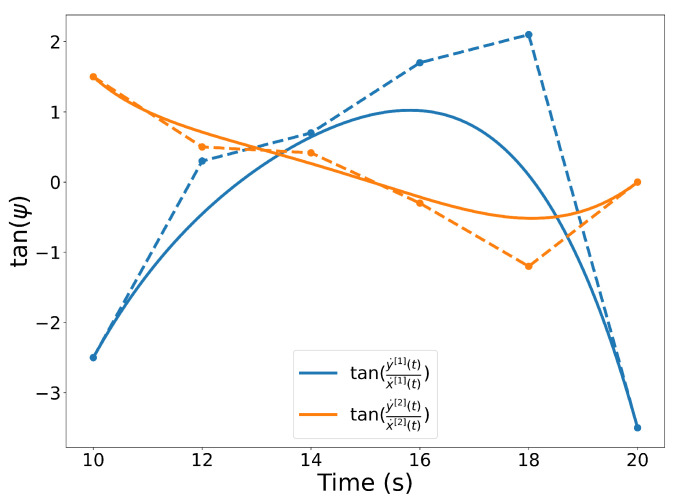
Illustration of the quantity expressed in Equation (Equation 8) for trajectories $C^{\left[1\right]}\left(t\right)$ and $C^{\left[2\right]}\left(t\right)$.

**Figure 8 sensors-22-01869-f008:**
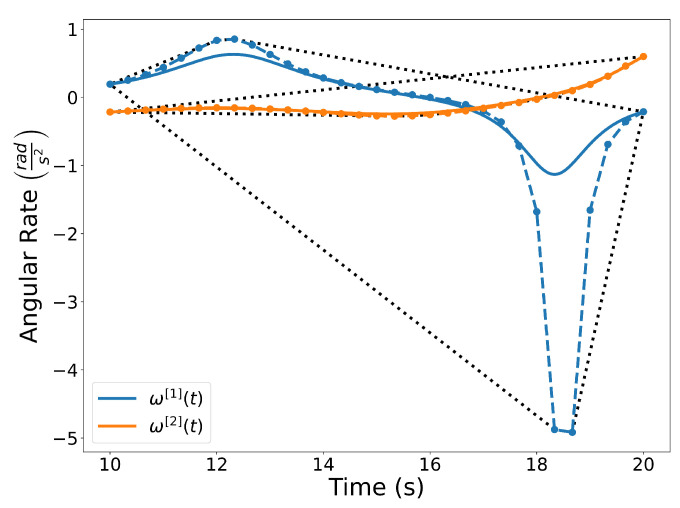
Angular rates of trajectories $C^{\left[1\right]}\left(t\right)$ and $C^{\left[2\right]}\left(t\right)$. Note that the angular rates are rational Bernstein polynomials.

**Figure 9 sensors-22-01869-f009:**
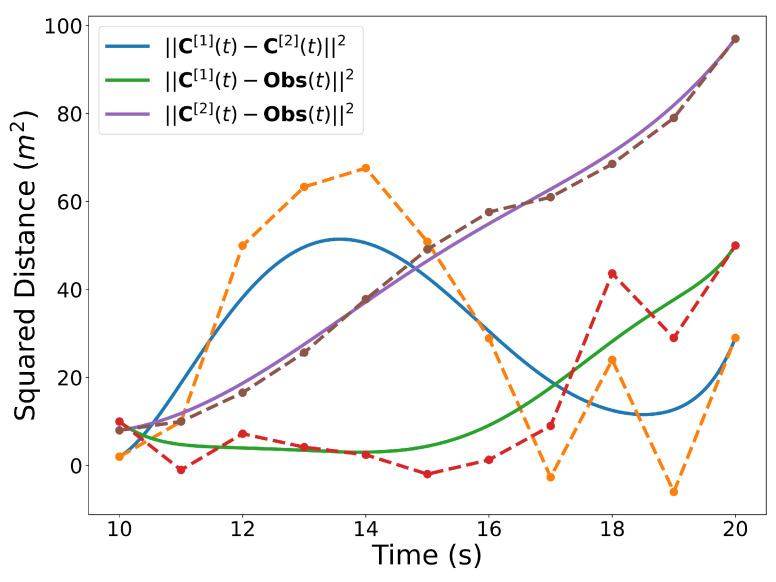
Squared distance between trajectories and the center of the circular obstacle.

**Figure 10 sensors-22-01869-f010:**
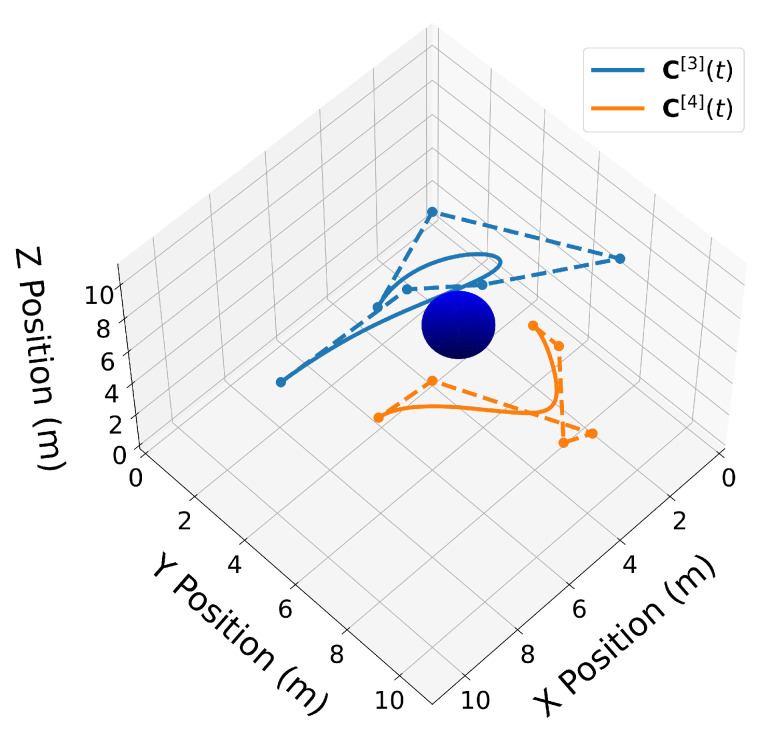
Two 3D Bernstein polynomial trajectories near a spherical obstacle. Trajectory $C^{\left[3\right]}\left(t\right)$ is drawn in blue and trajectory $C^{\left[4\right]}\left(t\right)$ is drawn in orange.

**Figure 11 sensors-22-01869-f011:**
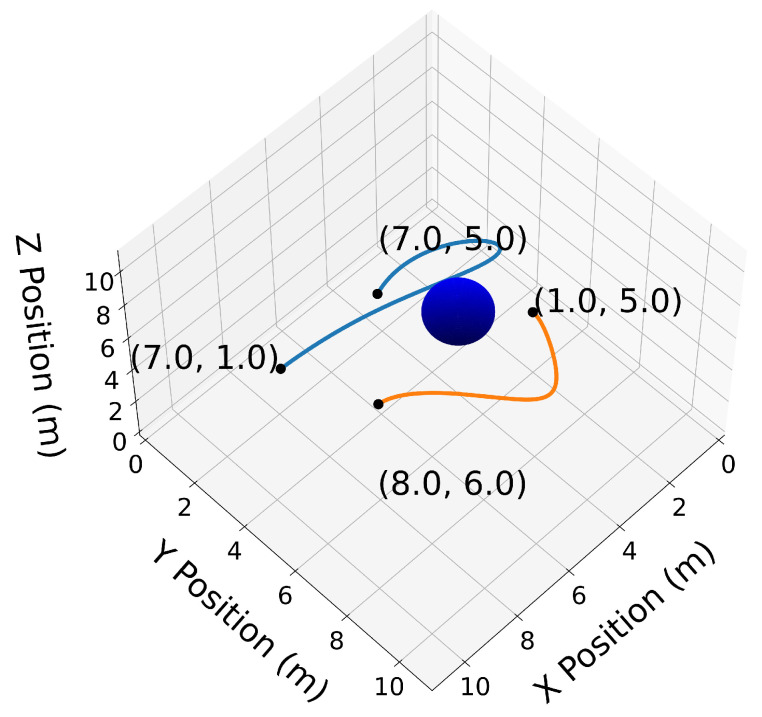
3D trajectories $C^{\left[3\right]}\left(t\right)$ and $C^{\left[4\right]}\left(t\right)$ with their endpoints highlighted.

**Figure 12 sensors-22-01869-f012:**
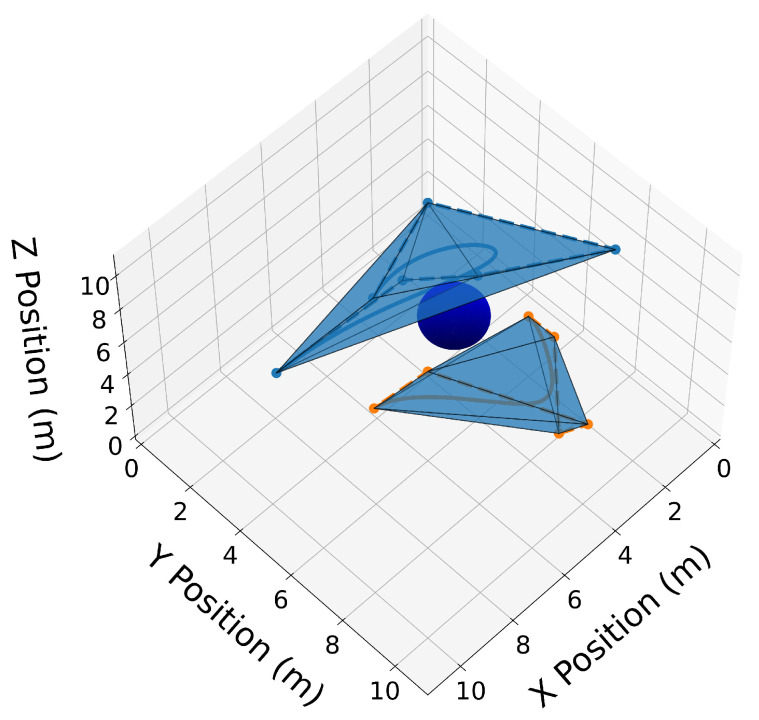
3D convex hulls drawn as transparent blue surfaces around the Bernstein coefficients of trajectories $C^{\left[3\right]}\left(t\right)$ and $C^{\left[4\right]}\left(t\right)$.

**Figure 13 sensors-22-01869-f013:**
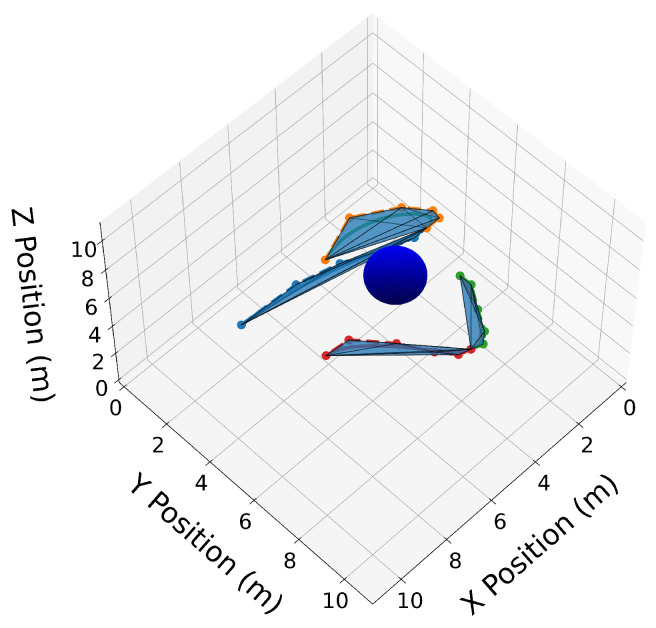
Trajectories $C^{\left[3\right]}\left(t\right)$ and $C^{\left[4\right]}\left(t\right)$ split at $t_{\mathrm{div}}=15$ s. Convex hulls are drawn around the Bernstein coefficients of the new split trajectories.

**Figure 14 sensors-22-01869-f014:**
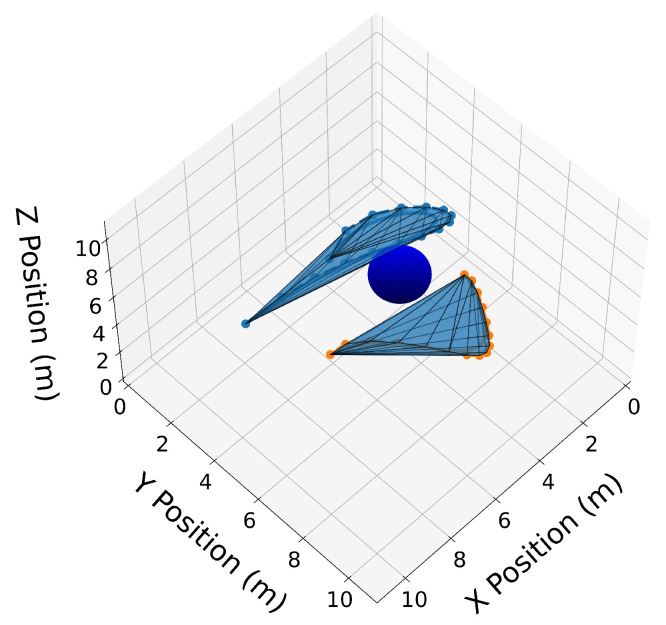
Convex hull drawn around the elevated Bernstein coefficients of trajectories $C^{\left[3\right]}\left(t\right)$ and $C^{\left[4\right]}\left(t\right)$.

**Figure 15 sensors-22-01869-f015:**
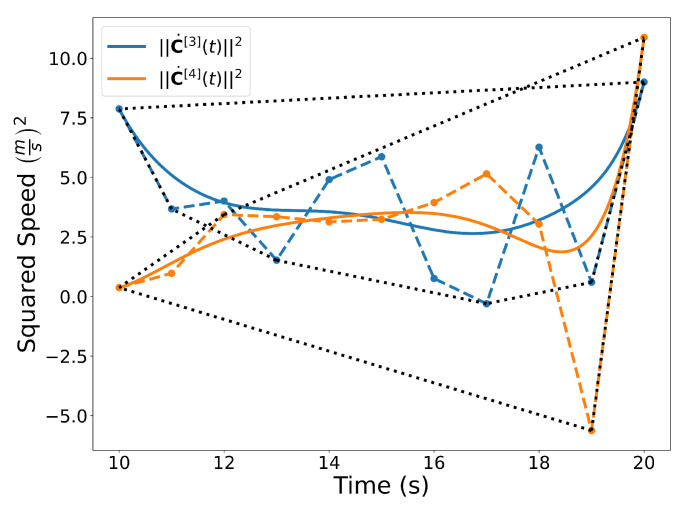
Squared speed of the trajectories $C^{\left[3\right]}\left(t\right)$ and $C^{\left[4\right]}\left(t\right)$. A convex hull is drawn around the Bernstein coefficients. Note that even though the coefficients may be negative, the actual curve is not.

**Figure 16 sensors-22-01869-f016:**
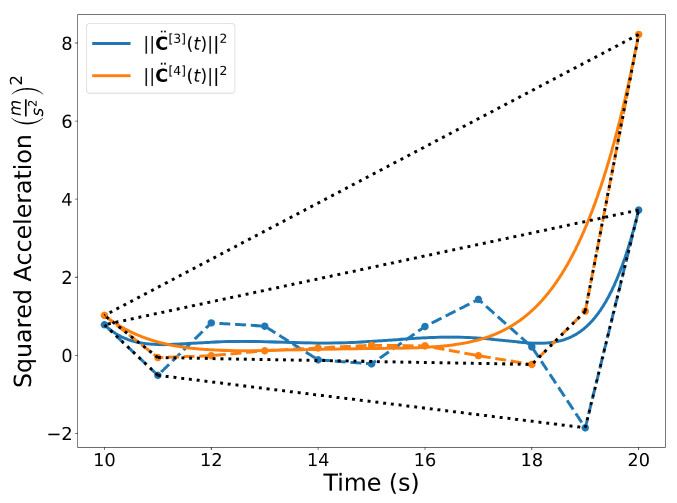
Squared acceleration of the trajectories $C^{\left[3\right]}\left(t\right)$ and $C^{\left[4\right]}\left(t\right)$ with corresponding convex hulls.

**Figure 17 sensors-22-01869-f017:**
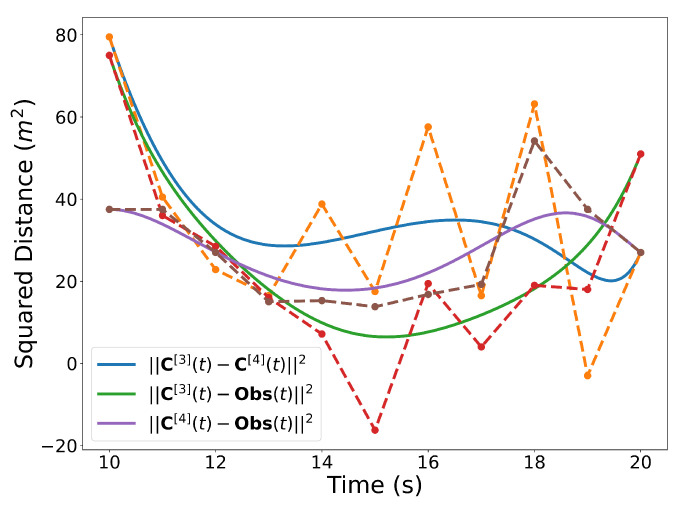
Squared distances between the trajectories and then center of the spherical obstacle.

**Figure 18 sensors-22-01869-f018:**
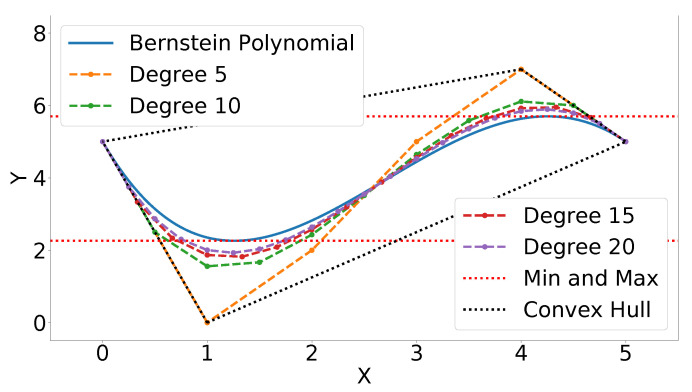
Bounds for Bernstein polynomials. The solid blue line is the Bernstein polynomial, the dashed lines connect the coefficients of each different order, the black dotted line represents the convex hull of the 5th degree Bernstein polynomial, and the red dotted lines represent the actual extrema.

**Figure 19 sensors-22-01869-f019:**
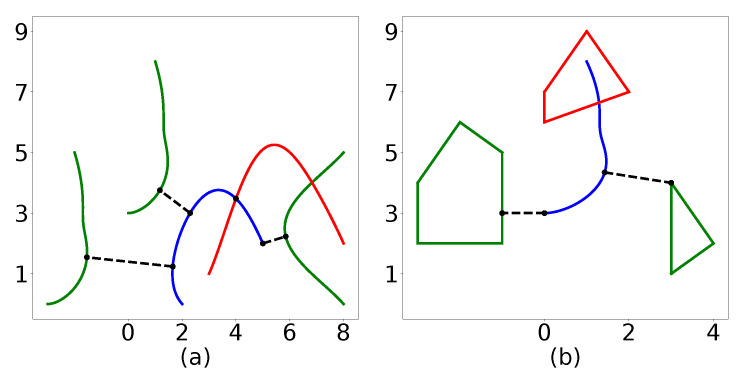
(**a**) Minimum distance between curves. (**b**) Minimum distance between a curve and a polygon. All distances are measured to the blue curve. A red curve or polygon indicates that a collision exists.

**Figure 20 sensors-22-01869-f020:**
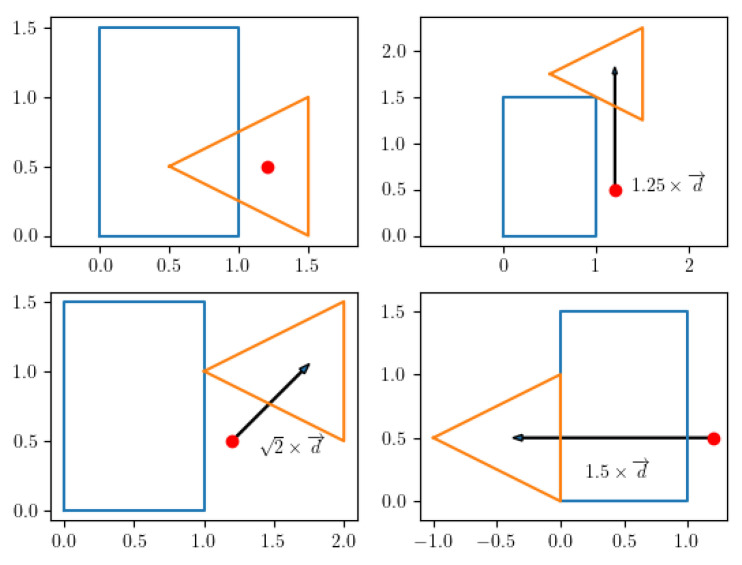
Illustration of penetration.

**Figure 21 sensors-22-01869-f021:**
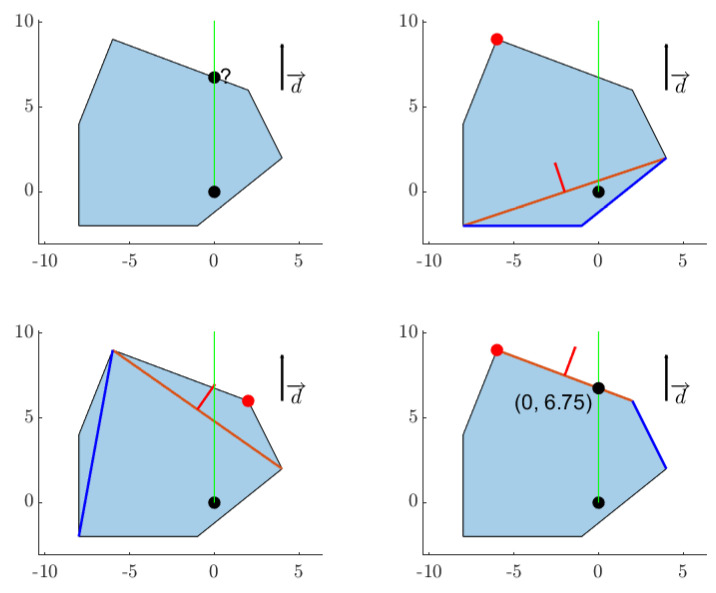
Iteration of the DEPA algorithm.

**Figure 22 sensors-22-01869-f022:**
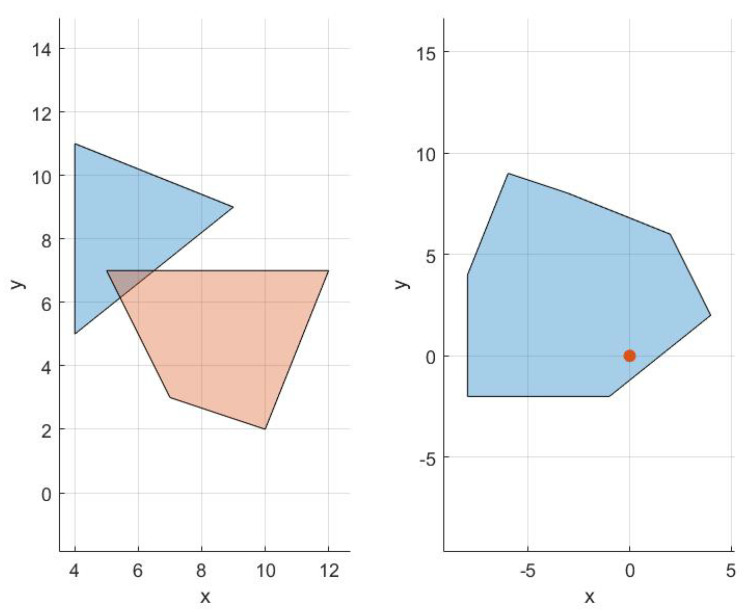
Two intersecting polygons and resulting Minkowski Difference.

**Figure 23 sensors-22-01869-f023:**
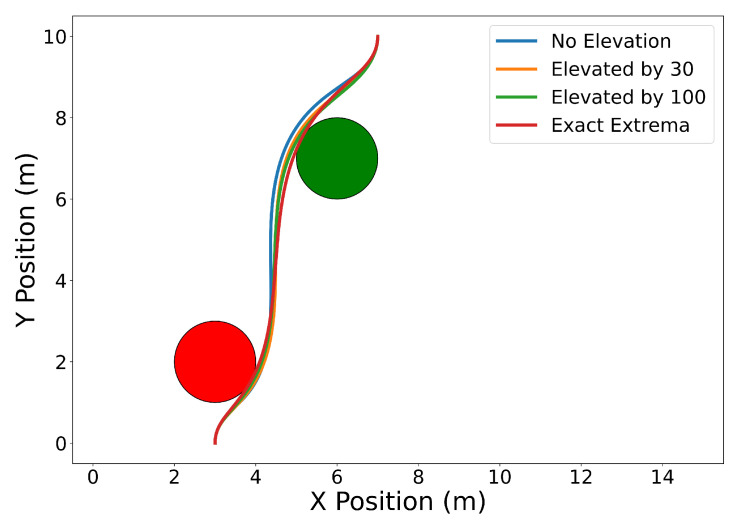
Time optimal trajectory for vehicle with initial and final speeds and headings, maximum speed, maximum angular rate, and maximum safe distance constraints ranging from least to most conservative distance estimates.

**Figure 24 sensors-22-01869-f024:**
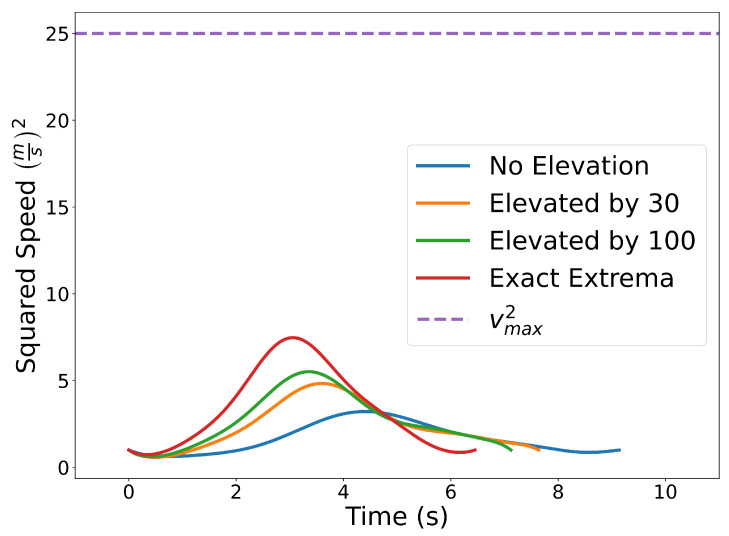
Plot of the squared speed constraints for each separate trial.

**Figure 25 sensors-22-01869-f025:**
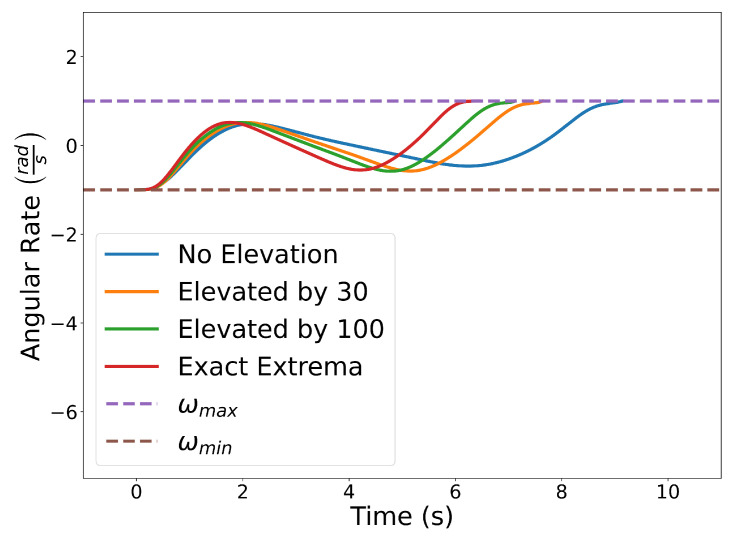
Plot of the angular rate constraints for each separate trial.

**Figure 26 sensors-22-01869-f026:**
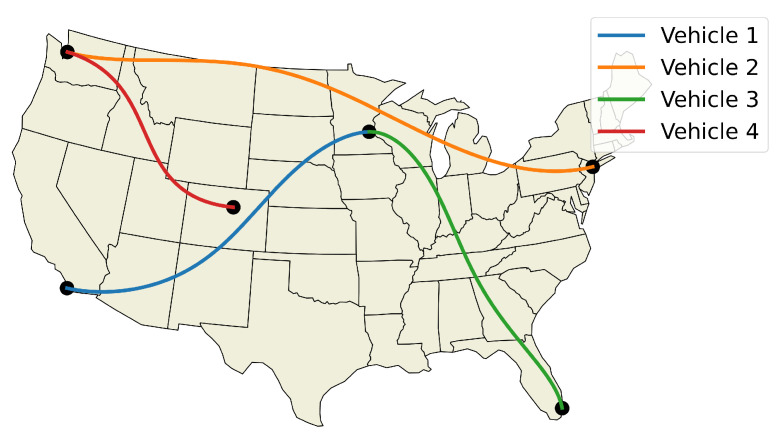
Commercial flight trajectories between major US cities.

**Figure 27 sensors-22-01869-f027:**
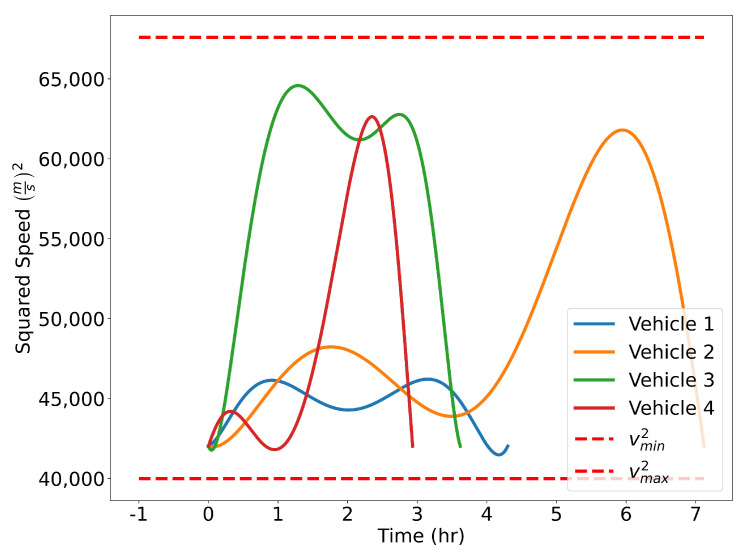
Verifying speed constraints for the Air Traffic Control example.

**Figure 28 sensors-22-01869-f028:**
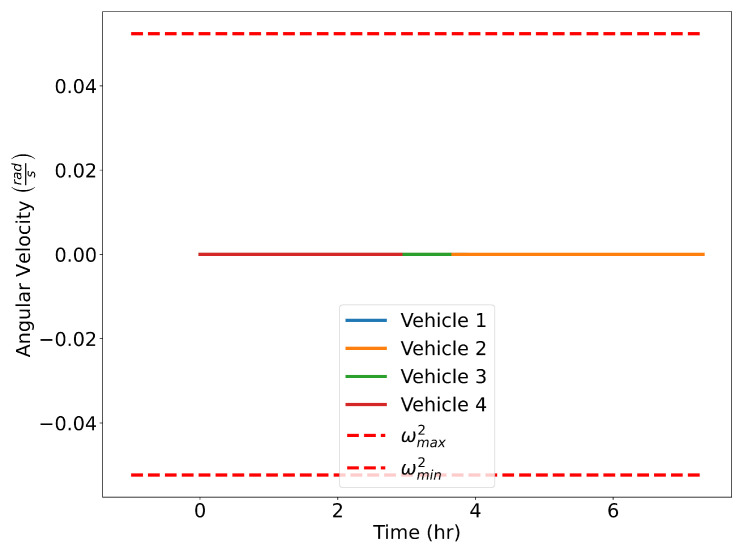
Verifying angular rate constraints for the Air Traffic Control example.

**Figure 29 sensors-22-01869-f029:**
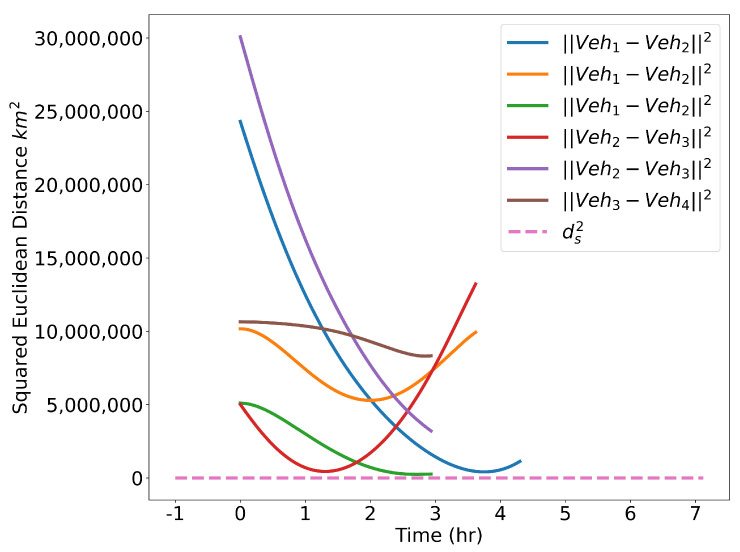
Verifying minimum safe distance constraints for the Air Traffic Control example.

**Figure 30 sensors-22-01869-f030:**
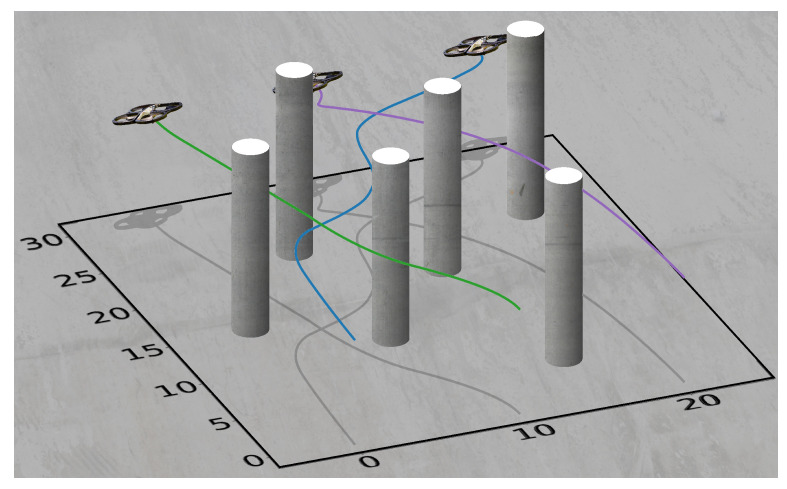
Aerial vehicles navigating a cluttered environment.

**Figure 31 sensors-22-01869-f031:**
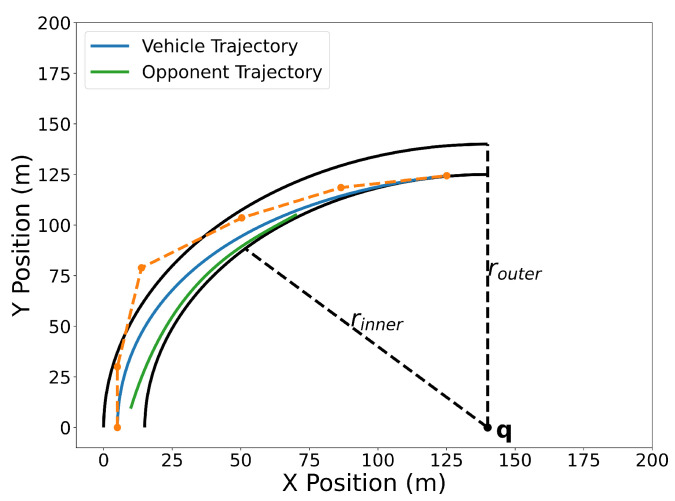
Overtaking trajectory. The track follows a quarter circle whose center point is at q. The inner track line has a radius of $r_{inner}$ and the outer track line has a radius of $r_{outer}$.

**Figure 32 sensors-22-01869-f032:**
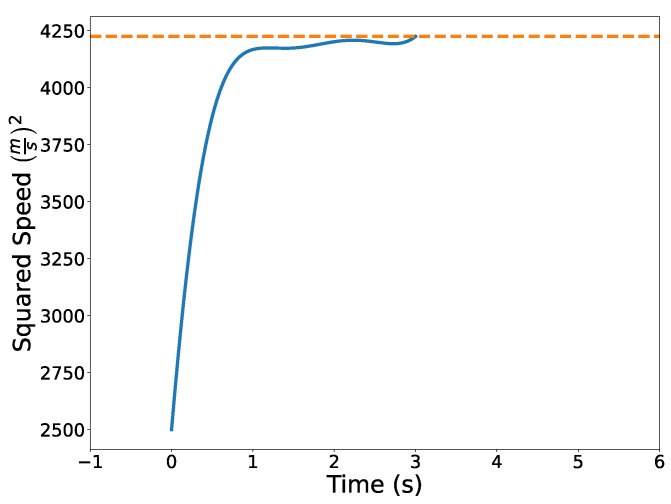
Squared speed profile of the vehicle.

**Figure 33 sensors-22-01869-f033:**
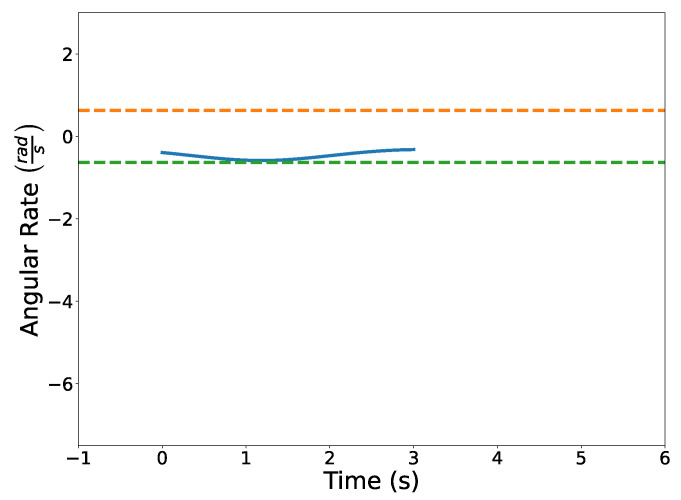
Angular rate profile of the vehicle.

**Figure 34 sensors-22-01869-f034:**
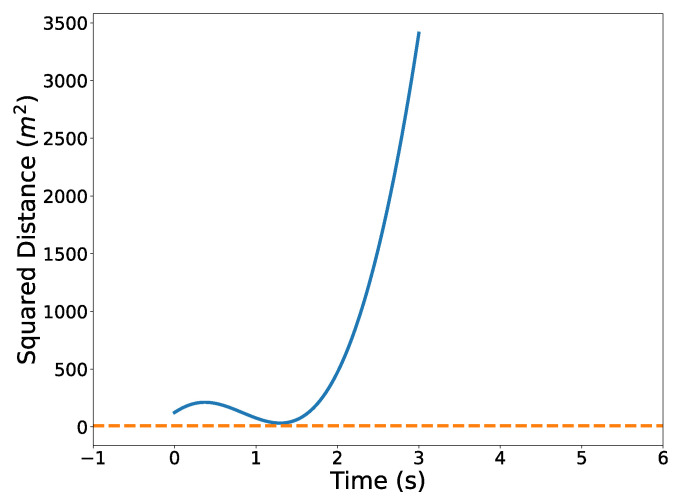
Squared distance between vehicle and opponent.

**Figure 35 sensors-22-01869-f035:**
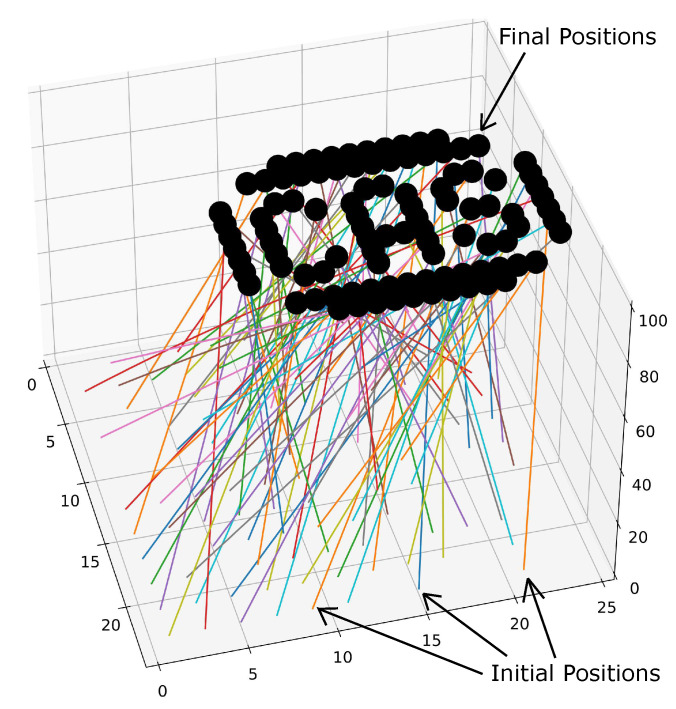
101 vehicles spelling out CAS using the centralized method.

**Figure 36 sensors-22-01869-f036:**
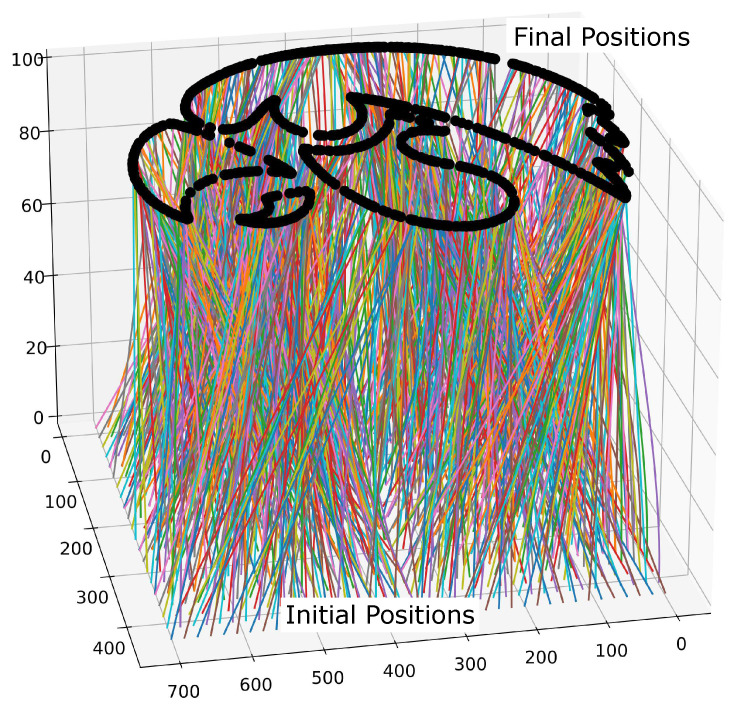
Trajectories for 1000 aerial vehicles with initial and final position and minimum safety distance constraints.

**Figure 37 sensors-22-01869-f037:**
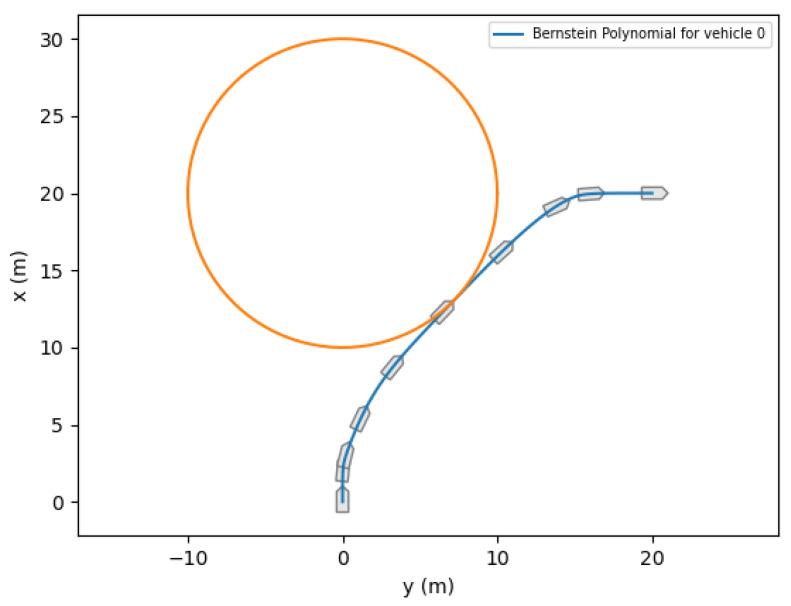
Medusa Example.

**Figure 38 sensors-22-01869-f038:**
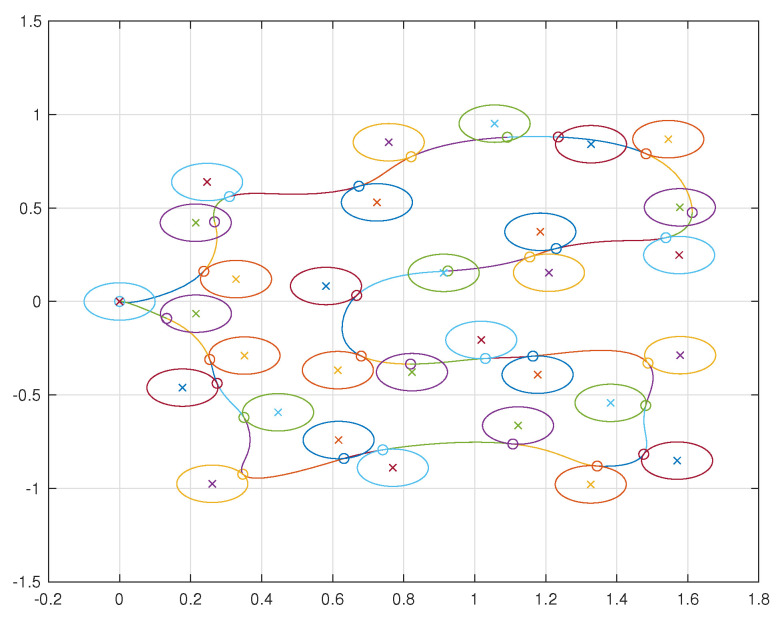
Trajectory of a single agent visiting 30 neighborhoods. The circles with an X in their center represent the neighborhoods and the circles along the vehicle’s trajectory represent the points at which the vehicle makes its delivery.

**Figure 39 sensors-22-01869-f039:**
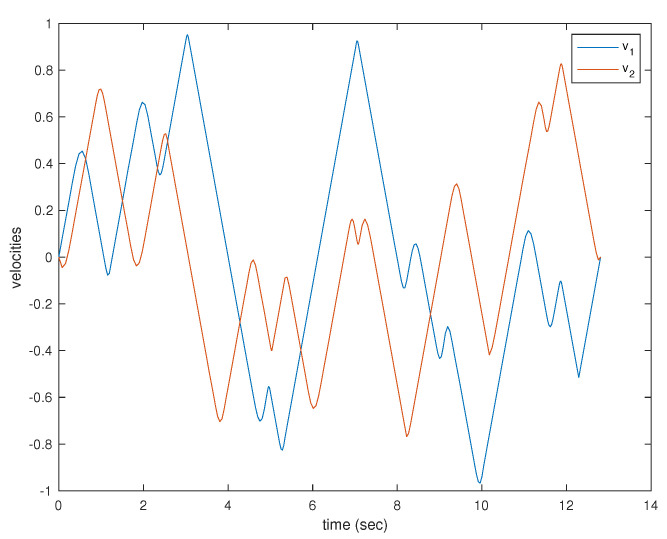
Velocity components of a single agent visiting 30 neighborhoods.

**Figure 40 sensors-22-01869-f040:**
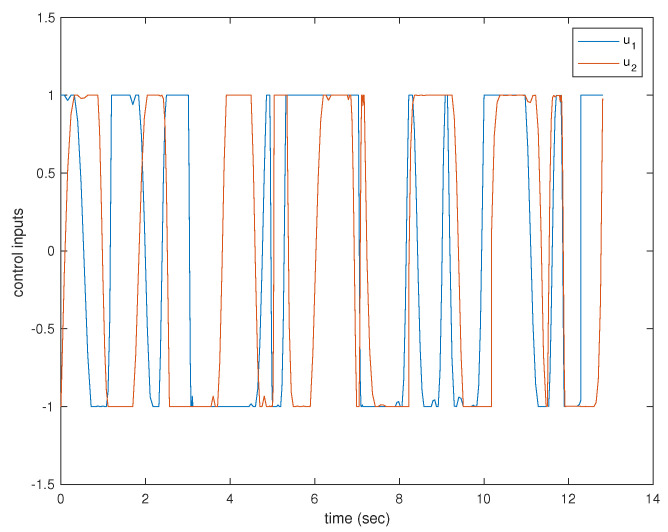
Acceleration components of a single agent visiting 30 neighborhoods.

**Figure 41 sensors-22-01869-f041:**
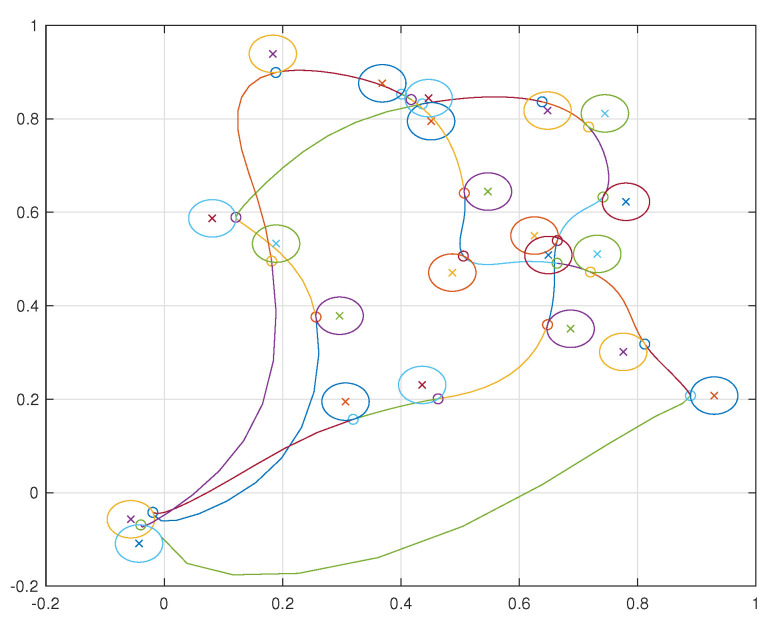
Two agents visiting ten neighborhoods each.

**Figure 42 sensors-22-01869-f042:**
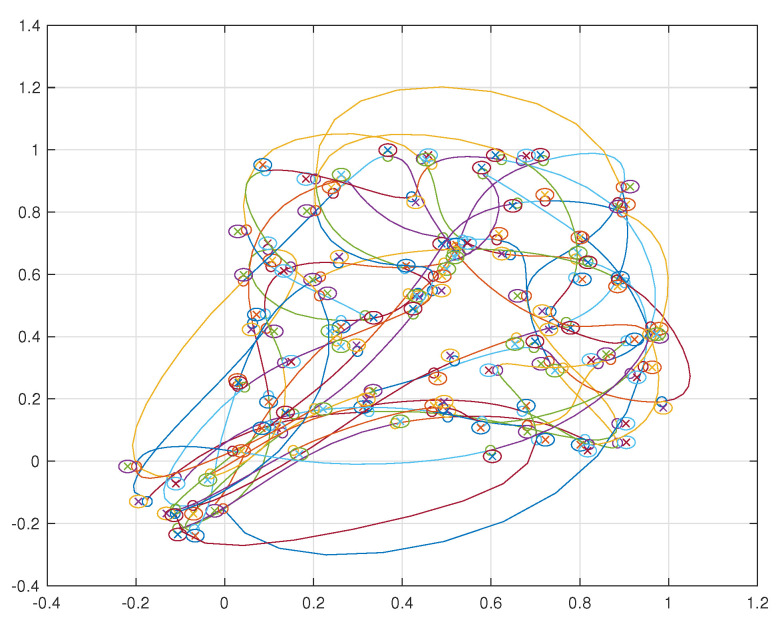
Ten agents visiting ten neighborhoods each.

## Data Availability

Not applicable.

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
