# Peer review of "Bernstein Polynomial-Based Method for Solving Optimal Trajectory Generation Problems"

_sensors, 2022, doi:10.3390/s22051869_

Round 1

Reviewer 1 Report

The authors propose a trajectory generation method and develop an open-source toolbox BeBOT (Bernstein/Bézier Optimal Trajectories). Several Numerical Examples demonstrate the effectiveness of the work. This work and open-source tools may arouse the interest of researchers in the field of motion planning. After carefully reading the manuscript and the authors’ conference paper[35], the reviewer would like to make the following suggestions.

  1. The authors mentioned, "By parameterizing the trajectories as Bernstein polynomials, this problem can then be transcribed into a nonlinear programming problem…"

In the opinion of the reviewer, there are many works use Bernstein polynomials for trajectory optimization, such as the use of convex hull properties and derivatives properties and so on. So what is the author's contribution in this field? The contributions and innovations of the manuscript should be clearly listed, preferably in the form of 1.2.3... at the end of the introduction part.

  1. The authors mentioned, "Our method for trajectory generation builds upon [32–34], where Bernstein polynomials were introduced as a tool to approximate the solutions of nonlinear optimal control problems with provable convergence guarantees."

The authors should make it clear, what are the improvements of the proposed method in this manuscript compared with the above-mentioned method [32–34]? How is the proposed method different from these methods?

  1. The authors mentioned, "This paper extends the results initially presented in [35] … we extend previous work by exploiting properties and proposing algorithms for both Bernstein polynomials and rational Bernstein polynomials."

The authors should clearly indicate the beneficial effects of these improvements and supplement comparative experiments to illustrate the beneficial effects of the above improvements is needed. What's more the author needs to list more clearly the improvements and extensions of this manuscript relative to the conference paper [35].

  1. The authors show the application of the proposed method in a series of Numerical Examples but lack comparative experiments with other methods in the same environment. The reviewer suggests that authors add comparative experiments with other methods, such as the method or existing toolkits mentioned in the introduction, and the method[32–34] which is mentioned above.

 In the same environment, how does the proposed method compare to other methods in terms of trajectory quality and success rate? Comparative experiments with other methods can also better demonstrate the performance of the proposed method.

  1. The section3 is missing in line 147.
  2. The manuscript is a bit long, and the introductory content can be shown in the appendix.
  3. References [36] and [46] do not have correct website links.

References in the manuscript:

  1. Cichella, V.; Kaminer, I.; Walton, C.; Hovakimyan, N. Optimal Motion Planning for Differentially Flat Systems Using Bernstein Approximation. IEEE Control Systems Letters 2018, 2, 181–186. doi:10.1109/LCSYS.2017.2778313.
  2. Cichella, V.; Kaminer, I.; Walton, C.; Hovakimyan, N.; Pascoal, A. Bernstein approximation of optimal control problems. arXiv preprint arXiv:1812.06132 2018.
  3. Cichella, V.; Kaminer, I.; Walton, C.; Hovakimyan, N.; Pascoal, A.M. Optimal Multi-Vehicle Motion Planning using Bernstein Approximants. IEEE Transactions on Automatic Control 2020.
  4. Kielas-Jensen, C.; Cichella, V. BeBOT: Bernstein polynomial toolkit for trajectory generation. 2019 IEEE/RSJ International Conference on Intelligent Robots and Systems (IROS). IEEE, 2019, pp. 3288–3293.
  5. Kielas-Jensen, C.; Cichella, V. BeBOT, 2019.

46. Kielas-Jensen, C.; Cichella, V. Trajectory Generation Using Bernstein Polynomials (Bézier Curves), 2019. Accessed: 2019-07-31.

Reviewer 2 Report

This article discusses the use of Bernstein polynomials to characterize vehicle trajectories, in order to use them in optimization algorithms with the goal of obtaining optimal plans. The authors show that, given the properties of these polynomials, an infinite-dimensional problem can be translated to a finite optimization task. Improved results can be obtained by elevating the degree of the Bernstein polynomial, and many existing nonlinear optimization algorithms can be leveraged.

The manuscript is well written, with several examples of applications and enough figures to clearly understand the proposed methods.

However, the contribution of this article is unclear, since most of the contents have already been published in reference [34] by, mostly, the same group of authors. The newly added application examples do not contribute knowledge or techniques to what is already stated in [34]. The software toolbox introduced in this article is not a research contribution.

Unless the authors present a relevant modification or improvement over [34], this article cannot be accepted as it is for publication. The applications shown in this article are not very different from those in [34] either.

Reviewer 3 Report

Interesting paper. In future work it would be also interesting to experimentally compare the proposed algorithms with other algorithms. 

Reviewer 4 Report

Comments on the paper “Bernstein polynomial-based method for solving optimaltrajectory generation problems“

Content comments

  1. This paper presents an interesting proposal, but some points must be improved before the paper is approved.
  2. The proposed approach is presented in [35]. Therefore, this research requires explicit presentation of the novelty aspects and the presentation of the numerical results showing the superiority of these novelty aspects. More details should be furnished.
  3. The proposed approach deals with the path planning for the autonomous robots. This approach is quite resource consuming, maybe is better to perform path planning by the iterative technique? Additional analysis concerning this aspect is optional. More details should be furnished.
  4. The authors state that they proposed an approach “ for solving optimal trajectory generation problems“. But the results presented in Fig. 3-7 need the comments with respect to this topic. x More details should be furnished.
  5. The authors state “The main motivation for this approach is that Bernstein polynomials possess favorable geometric properties and yield computationally efficient algorithms that enable a trajectory planner to efficiently evaluate and enforce constraints along the vehicles’ trajectories, including maximum speed and angular rates as well as minimum distance between trajectories and between the vehicles and obstacles“. Therefore, the governing motion equations must be included in the proposed mathematical models. More details should be furnished
  6. The Fig. 37-38 must be redesigned, because the presented versions are noninformative.
  7. The conclusions must be more closely directed to the novelty aspects of the paper.
  8. During the last decade a lot of the research is performed on this topic. Therefore, additional references could be also included, such as
    • Vasconcelos, J.V.R.; Brandão, A.S.; Sarcinelli-Filho, M. Real-Time Path Planning for Strategic Missions.  Sci.202010, 7773. https://doi.org/10.3390/app10217773
    • Semenas, R.; Bausys, R., Zavadskas E.K. A novel environment exploration strategy by m-generalised q-neutrosophic WASPAS. Studies in informatics and control 2021, vol. 30, iss. 3, p. 19-28. https://doi.org/10.24846/v30i3y202102

Round 2

Reviewer 1 Report

The authors responded to the reviewers' comments clearly, and the manuscript has been sufficiently improved to warrant publication in Sensors. Acceptance of the article is recommended

Reviewer 2 Report

The authors have now clearly differentiated their article from their prior work. While this work does not represent a major contribution, it does introduce novel algorithms that exploit the Bernstein polynomials for optimal trajectory generation. Coupled with the examples provided by the authors, this article becomes an asset for the practitioner. 

Reviewer 4 Report

The manuscript can be published.